# Understanding Hallucinations in Diffusion Models through Mode Interpolation

**Sumukh K Aithal**[1]     **Pratyush Maini**[1,2]     **Zachary C. Lipton**[1]     **J. Zico Kolter**[1]
Carnegie Mellon University[1]     DatologyAI[2]
{saithal, pratyus2, zlipton, zkolter}@cs.cmu.edu

## Abstract

Colloquially speaking, image generation models based upon diffusion processes are frequently said to exhibit "hallucinations"—samples that could never occur in the training data. But where do such hallucinations come from? In this paper, we study a particular failure mode in diffusion models, which we term *mode interpolation*. Specifically, we find that diffusion models smoothly "interpolate" between nearby data modes in the training set to generate samples that are completely outside the support of the original training distribution; this phenomenon leads diffusion models to generate artifacts that never existed in real data (i.e., hallucinations). We systematically study the reasons for, and the manifestation of this phenomenon. Through experiments on 1D and 2D Gaussians, we show how a discontinuous loss landscape in the diffusion model's decoder leads to a region where any smooth approximation will cause such hallucinations. Through experiments on artificial datasets with various shapes, we show how hallucination leads to the generation of combinations of shapes that never existed. We extend the validity of mode interpolation in real-world datasets by explaining the unexpected generation of images with additional or missing fingers similar to those produced by popular text-to-image generative models. Finally, we show that diffusion models in fact *know* when they go out of support and hallucinate. This is captured by the high variance in the trajectory of the generated sample towards the final few backward sampling steps. Using a simple metric to capture this variance, we can remove over 95% of hallucinations at generation time while retaining 96% of in-support samples in the synthetic datasets. We conclude our exploration by showing the implications of such hallucination (and its removal) on the collapse (and stabilization) of recursive training on synthetic data with experiments on MNIST and a 2D Gaussians dataset. We release our code at https://github.com/locuslab/diffusion-model-hallucination.

## 1 Introduction

The high quality and diversity of images generated by diffusion models [15, 38] have made them the de facto standard generative models across various tasks including video generation [6], image inpainting [24], image super-resolution [11], data augmentation [44], and others. As a result of their uptake, large volumes of synthetic data are rapidly proliferating on the internet. The next generation of generative models will likely be exposed to many machine-generated instances during their training, making it crucial to understand ways in which diffusion models fail to model the true underlying data distribution. Like other generative model families, much research has been done to understand the failure modes of diffusion models as well. Past works have identified, and attempted to explain and remedy failures such as, training instabilities [17], memorization [7, 39] and inaccurate modeling of objects such as hands and legs [4, 23, 28].

38th Conference on Neural Information Processing Systems (NeurIPS 2024).

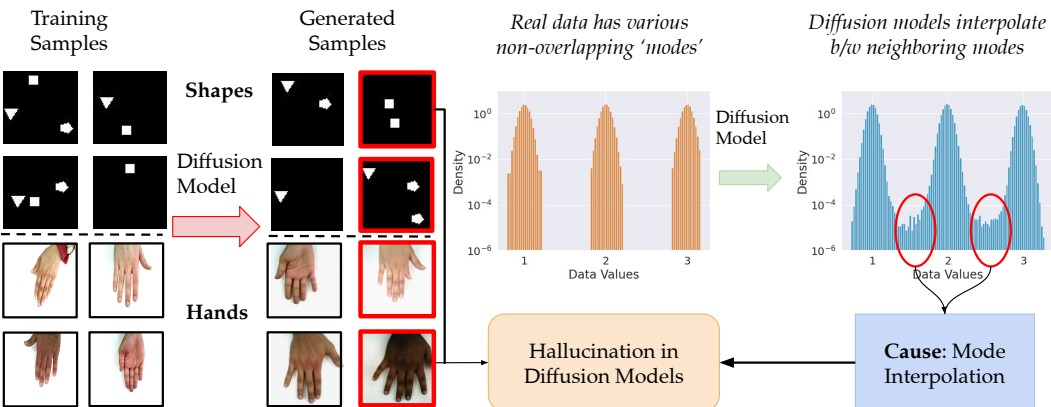

Figure 1: **Hallucinations in Diffusion Models**: Original Dataset (Left) & Generated Dataset (Right). (Top) The original dataset consists of 64x64 images divided into three columns, each containing a triangle, square, or pentagon with a 0.5 probability of the shape being present. Each shape appears at most once per image. The generated dataset created using an unconditional DDPM includes some samples (*hallucinations*) with multiple occurrences of the same shape that is unseen in the original dataset. (Bottom) We also train a ADM [29] on a dataset of high-quality images of human hands and show that the diffusion model generates hallucinated images of hands with additional fingers.

In this work, we formalize and study a particular failure mode of diffusion models that we call hallucination—a phenomenon where diffusion models generate samples that lie completely out of the support of the training distribution of the model. As a contemporary example, hallucinations manifest in large generative models like StableDiffusion [33] in the form of hands with extra (or missing) fingers or limbs. We begin our investigation with a surprising observation that an unconditional diffusion model trained on a distribution of simple shapes, generates images with combinations of shapes (or artifacts) that never existed in the original training distribution (Figure 1). While extensive research on generative models has focused on the phenomenon of 'mode collapse' [48], which leads to a loss of diversity in the sampled distribution, such studies often overlook the complex nature of real data which typically comprise multiple distinct modes on a complex data manifold, and the effects of their mutual interactions are thus neglected. In our work, we explain hallucinations by introducing a novel phenomenon we term 'mode interpolation' that considers this mutual interaction.

To understand the cause of these hallucinations and their relationship to mode interpolation, we construct simplified 1-d and 2-d mixture of Gaussian setups and train diffusion models on them (§ 4). We observe that when the true data distribution occurs in disjoint modes, diffusion models are unable to model a true approximation of the underlying distribution. This is because there exist 'step functions' between different modes, but the score function learned by the DDPM is a smooth approximation of the same, leading to interpolation between the nearest modes, even when these interpolated values are entirely absent from the training data. Moreover, we observation that hallucinated samples usually have very high variance towards the end of their trajectory during the reverse diffusion process. Based on this observation, we use the trajectory variance during sampling as a metric to detect hallucinations (§ 5), and show that diffusion models usually 'know' when they hallucinate, allowing detection with sensitivity and specificity > 0.92 in our experiments.

We explore mode interpolation as a potential explanation for the common failure of large-scale generative models, to accurately generate human hands. To demonstrate this concretely, we trained a diffusion model on a dataset of high-quality hand images and observed that it generated hands with additional fingers. We then applied our proposed metric to effectively detect these hallucinated generations. Finally, we study the implications of this phenomenon in recursive generative model retraining where we train generative models on their own output (§ 6). Recently, recursive training and its downsides in model collapse have garnered a lot of attention in both language and diffusion modeling literature [2, 3, 5, 9]. We observe that the proposed detection mechanism is able to mitigate the model collapse during recursive training on 2D Grid of Gaussians, Shapes and MNIST dataset.

## 1.1 Hallucination in Diffusion Models

Before formalizing our notions and definitions in § 3, let us first consolidate the observation that has been loosely labeled as 'hallucination' until now. To illustrate this phenomenon, we design a synthetic dataset called SIMPLE SHAPES, and train a diffusion model to learn its distribution.

**SIMPLE SHAPES Setup.** Consider a dataset consisting of black and white images that contain three shapes: triangle, square, and pentagon. Each image in the dataset is 64x64 pixels in size and divided into three (implied) columns. The first, second, and third columns contain a triangle, square, and pentagon, respectively. Each column has a 0.5 probability of containing the corresponding shape. A representation of this setup is shown in Fig 1. It is important to note that in this data generation pipeline, each shape is present at most once in each image.

**Observation.** We train an unconditional Denoising Diffusion Probabilistic Model (DDPM) [15] on this toy dataset with $T = 1000$ timesteps. We observe that the DDPM generates a small fraction of images that are never observed in the training dataset, nor a part of the 'support' of the data generation pipeline. Specifically, the model generates some images that contain two occurrences of the same shape, as shown in Fig 1. Furthermore, when the model is iteratively trained on its own sampled data, the fraction of these occurrences increases significantly as the generation process progresses.

Inspired by these observations and their implications, we will perform experiments through the rest of this work to formalize what we mean by hallucinations (§ 3), why do they occur (§ 4), how can we mitigate them (§ 5), and what are their implications for real-world datasets (§ 6).

## 2 Related Work

**Diffusion Models.** Diffusion models [15, 38, 43] are a class of generative models characterized by a forward process and a reverse process. In the forward process, noise is incrementally added to an image over time steps, ultimately converting the data into noise. The reverse process learns to denoise the image using a neural network essentially learning to convert noise to data. Diffusion models have various interpretations. Score-based generative modeling [41, 42] and DDPMs [15] are closely related, with [43] proposing a unified framework using stochastic differential equations (SDEs) that generalizes both Score Matching with Langevin Dynamics (SMLD) [43] and DDPM. In this framework, the forward process is a SDE with a continuous-time generalization instead of discrete timesteps and the reverse process is also an SDE that can be solved using a numerical solver. Another perspective is to view diffusion models as hierarchical Variational Autoencoders (VAEs) [25]. Recent research [19] suggests that diffusion models learn the optimal transport map between Gaussian distribution and data distribution. In this paper, we discover a surprising phenomenon in diffusion which we coin mode interpolation.

**Recursive Generative Model Training.** Recent works [2, 3, 26, 27, 37] demonstrated that iteratively training the generative models on their own output (i.e recursive training) leads to model collapse. The model collapse can happen in two ways: either all samples collapse to a single mode (low diversity) or the model generates very low fidelity, unrealistic images (low sample quality). This has been shown in the visual domain with StyleGAN2 and diffusion models [2, 3], as well as in the text domain with Large Language Models (LLMs) [5, 9, 37]. The current solution to mitigate this collapse is to include a fraction of real data in the training loop at all the generations [2, 3]. Theoretical results have also proved that super-quadratic number of synthetic samples are necessary to prevent model collapse [10] in the absence of support from real data. A concurrent work [12] studied the setup of data accumulation in recursive training where data from previous iterations of generative models together with real data are accumulated over time. The authors conclude that data accumulation (including real data) can avoid model collapse in various settings including language modeling and image data.

Past works have only studied the collapse of the generative model to the mode of the existing distribution. Through some controlled experiments, we study the interaction between different modes (a mode can be a class) or novel modes being developed in the generative models. This provides novel insights into the reasons behind the collapse of generative models during recursive training.

**Failure Modes of Diffusion Models.** One of the common failure modes of diffusion models is the generation of images where the hands and legs appear distorted or deformed which is commonly observed in Stable Diffusion [33] and Sora [6]. Diffusion models also fail to learn rare concepts [35]

which have less than 10k samples in the training set. Various other failure modes including ignoring spatial relationships or confusing attributes have been discussed in [4, 23].

**Hallucination in Language Models.** Hallucination in LLMs [46, 47] is a huge barrier to the deployment of LLMs in safety-critical systems. The LLMs may provide a factually incorrect output or incorrectly follow the instructions or be logically wrong. A simple example is that LLMs can generate new facts when asked to summarize a block of text (input-conflicting hallucination) [47]. Current hallucination mitigation techniques in LLMs include factual data enhancement [13], retrieval augmentation [32] among other methods. Given the widespread adoption of image generation models, we argue that hallucination in diffusion models must also be studied carefully to identify its causes and mitigate it.

## 3 Definitions and Preliminaries

Let $q(x)$ be the real data distribution. We define a forward process where Gaussian noise is iteratively added at each timestep for a total of $T$ timesteps. Let $x_0 \sim q(x)$, and $x_t$ be the perturbed (noisy) sample after adding $t$ timesteps of noise. The noise schedule is defined by $\beta_t \in (0, 1)$, which represents the variance of Gaussian (added noise) at time $t$. For large enough $T$, $x_T \sim \mathcal{N}(0, \mathbf{I})$

$$q(\mathbf{x}_t|\mathbf{x}_{t-1}) = \mathcal{N}(\sqrt{1-\beta_t}\mathbf{x}_{t-1}, \beta_t\mathbf{I}); \qquad q(\mathbf{x}_{1:T}|\mathbf{x}_0) = \prod_{t=1}^{T} q(\mathbf{x}_t|\mathbf{x}_{t-1}) \tag{1}$$

In the forward diffusion process, we can directly sample $x_t$ at any time step using the closed form $q(\mathbf{x}_t|\mathbf{x}_0) = \mathcal{N}(\mathbf{x}_t; \sqrt{\bar{\alpha}_t}\mathbf{x}_0, (1-\bar{\alpha}_t)\mathbf{I})$ where $\alpha_t = 1 - \beta_t$ and $\bar{\alpha}_t = \prod_{j=1}^{t} \alpha_j$.

The reverse diffusion process aims to learn the process of denoising i.e, learning $p_\theta(x_{t-1}|x_t)$ using a model (such as a neural network) with $\theta$ as the learnable parameters.

$$p_\theta(\mathbf{x}_{0:T}) = p(\mathbf{x}_T) \prod_{t=1}^{T} p_\theta(\mathbf{x}_{t-1}|\mathbf{x}_t); \qquad p_\theta(\mathbf{x}_{t-1}|\mathbf{x}_t) = \mathcal{N}(\mathbf{x}_{t-1}; \mu_\theta(\mathbf{x}_t, t), \Sigma_\theta(\mathbf{x}_t, t)) \tag{2}$$

The mean can be derived as $\mu_\theta(\mathbf{x}_t, t) = \frac{1}{\sqrt{\alpha_t}}\left(\mathbf{x}_t - \frac{1-\alpha_t}{\sqrt{1-\bar{\alpha}_t}}\epsilon_\theta(\mathbf{x}_t, t)\right)$ where $\epsilon_\theta(\mathbf{x}_t, t)$ is the predict noise at timestep $t$ using the neural network. The original DDPM is trained to predict the noise $\epsilon_t$ instead of $x_t$ and the variance $\Sigma_\theta(\mathbf{x}_t, t)$ is fixed and time-dependent. Since then, improved methods have learned the variance [29]. We define predicted $x_0$ as $\hat{x_0} = \frac{1}{\sqrt{\bar{\alpha}_t}}\left(\mathbf{x}_t - \sqrt{1-\bar{\alpha}_t}\epsilon_\theta(\mathbf{x}_t, t)\right)$

**Connections to Score Based Generative Models.** The score function $s(x)$ of a distribution $p(x)$ is the gradient of the log probability density function i.e, $\nabla_x \log p(x)$. The main premise of score-based generative modeling is to learn the score function of the data distribution given the samples from the same distribution. Once this score function is learned, annealed Langevin dynamics can be used to sample from the distribution using the formula $\mathbf{x}_{t+1} \leftarrow \mathbf{x}_t + \eta\nabla_\mathbf{x} \log p(\mathbf{x}) + \sqrt{2\eta}\mathbf{z}_t$, where $\eta$ is the step size and $z_t$ is sampled from standard normal. The score function can be obtained from the diffusion model using the equation $s_\theta(x_t, t) = -\frac{\epsilon_\theta(\mathbf{x}_t, t)}{\sqrt{1-\bar{\alpha}_t}}$ [45].

## 4 Understanding Mode Interpolation and Hallucination

In this section, we provide initial investigations into the central phenomenon of hallucinations in diffusion models. Formally, we consider a hallucination to be a generation from the model that lies entirely outside the support of the real data distribution (or, for distributions that theoretically have full support, in a region with negligible probability). That is, the $\epsilon$-Hallucination set $H_\epsilon(q)$

$$H_\epsilon(q) = \{x : q(x) \leq \epsilon\}, \tag{3}$$

where we typically take $\epsilon = 0$ or take $\epsilon$ to be vanishingly small (well beyond numerical precision). We similarly define the $\epsilon$-support set $S_\epsilon(q)$ to simply be the complement of the $\epsilon$-Hallucination set.

Mode interpolation occurs when a model generates samples that directly *interpolate* (in input space) between two samples in the $\epsilon$-support set, such that the interpolation lies in the $\epsilon$-Hallucination set. That, is for $x, y \in S_\epsilon(q)$ the model generates $\theta x + (1 - \theta)y \in H_\epsilon(q)$. The main argument of this

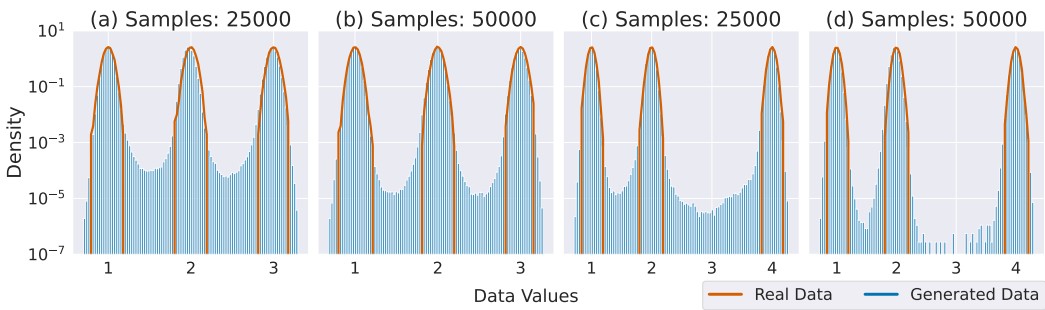

Figure 2: **Mode Interpolation in 1D GAUSSIAN**. The red curve indicates the PDF of the true data distribution $q(x)$, which is a mixture of 3 Gaussians (notice that the y-axis is in log-scale). In blue, we show a density histogram of the samples generated by a DDPM trained on varying number of samples from the true data distribution. For each histogram, we sampled 100 million examples from the diffusion model to observe the interpolated distribution. **(a,b)** show how the density of samples generated in the interpolated region reduces with an increase in the number of samples from the real distribution (used for training the DDPM). **(c,d)** show the impact of moving one of the modes (originally at $\mu = 3$) to $\mu = 4$. We see how the density of samples generated in the region between distant (but neighboring) modes is significantly lesser than that between nearby modes.

paper, shown through examples and numerical analysis of special cases, is that diffusion models frequently exhibit mode interpolation between "nearby" modes in the data distributions, and such interpolation leads to the generation of artifacts that did not exist in the original data (hallucinations).

### 4.1 1D GAUSSIAN Setup

We have already seen how hallucinations manifest in the SIMPLE SHAPES set-up (§ 1.1). To investigate hallucinations via mode interpolation, we begin with a synthetic toy dataset characterized by a mixture of 1D Gaussians given by: $p(x) = \frac{1}{3}\mathcal{N}(\mu_1, \sigma^2) + \frac{1}{3}\mathcal{N}(\mu_2, \sigma^2) + \frac{1}{3}\mathcal{N}(\mu_3, \sigma^2)$. For our initial experiments, we set $\mu_1 = 1, \mu_2 = 2, \mu_3 = 3$ and $\sigma = 0.05$. We sample 50k training points from this true distribution and train an unconditional DDPM using these samples with $T = 1000$ timesteps for $10,000$ epochs. Additional experimental details are present in the Appendix A.

We observe that diffusion models can generate samples that interpolate between the two nearest modes of the mixture of Gaussians (Figure 2). To clearly observe the distribution of these interpolated samples, we generated 100 million samples from the diffusion models. The probability of sampling from the interpolated regions (regions outside the support of the real data density, outlined in red) is non-zero, and decays with the distance from the modes. This region has nearly 0 probability mass of the true distribution, and no samples in this region occurred in the data used to train the DDPM.

The rate of mode interpolation depends primarily on three factors: **(i)** Number of training data points, **(ii)** variance of (and distance between) the distributions, and **(iii)** the number of sampling timesteps ($T$). As the number of training samples increases, we observe that the proportion of interpolated samples decreases. In this setup, the variance of $p(x)$ not only depends on $\sigma$ but also the distance between the modes i.e, $|\mu_1 - \mu_2|$ and $|\mu_2 - \mu_3|$. We run another experiment with $\mu_1 = 1, \mu_2 = 2$ and $\mu_3 = 4$. In this case, we observe that the frequency of samples between $\mu_2$ and $\mu_3$ is much lower than $\mu_1$ and $\mu_2$. The number of interpolated samples also decreases as the distance from the modes increases. The frequency of interpolated samples is also inversely proportional to the number of timesteps $T$. Additional experiments with varying Gaussian counts are in Appendix C.

### 4.2 2D GAUSSIAN Grid

The reduction in density of mode interpolation as two modes with $\mu = [2, 3]$ are moved apart calls for closer inspection into when and how diffusion models choose to interpolate between nearby modes. To investigate this, we make a toy dataset with a mixture of 25 Gaussians arranged in a two-dimensional square grid. A total of 100,000 samples are present in the training set. Similar to the 1D case, we observe interpolated samples between the two nearest modes of the Gaussian. Again, these samples have close to zero probability if sampled from the original distribution (Figure 3).

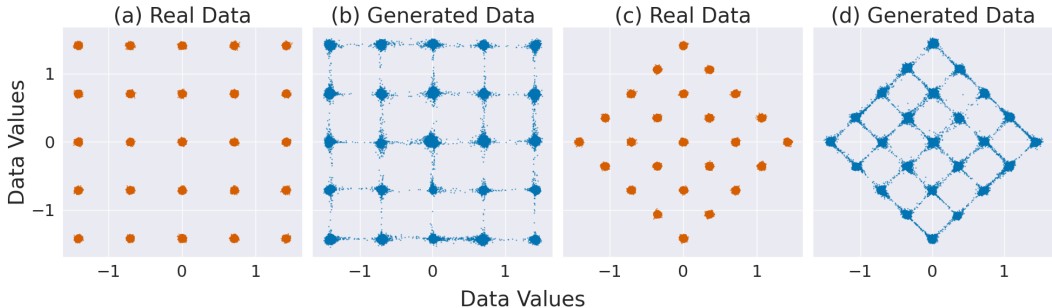

Figure 3: **Mode Interpolation in 2D GAUSSIAN.** The dataset consists of a mixture of 25 Gaussians arranged in a square grid, with a training set containing 100,000 samples. **(a,b)** The blue points represent samples generated by a DDPM, with visible density between the nearest modes of the original Gaussian mixture (in orange). These interpolated samples have near-zero probability in the original distribution. **(c,d)** We trained a DDPM on a rotated version of the dataset where the modes form a diamond shape. In this configuration, we see no interpolation along the x-axis, illustrating that diffusion models interpolate between nearest modes.

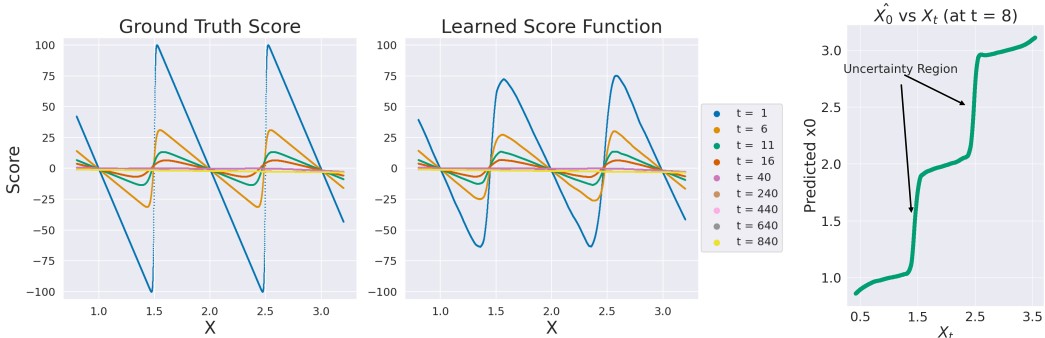

Figure 4: **Explaining Mode Interpolation via Learned Score Function.** The left panel shows the ground truth score function for a mixture of Gaussians across various timesteps, while the right panel illustrates the score function learned by the neural network. While the true score function exhibits sharp jumps that separate distinct modes (particularly in the initial time steps), the neural network approximates a smoother version.

We note that mode interpolation only happens between the nearest neighbors. To demonstrate this occurrence, we also train a DDPM on the rotated version of the dataset where the modes are arranged in the shape of a diamond (Figure 3.c,d). The mode interpolation can be more clearly observed in this setting. Interestingly, there appears to be no interpolation between modes along the x-axis, indicating that only the nearest modes are being interpolated. We believe this empirical observation of mode interpolation being confined to nearby modes will spark further investigation in future research.

## 4.3   What causes mode interpolation?

To understand the reason behind the observed mode interpolation, we analyze the score function learned by the model. The model learns to predict $\epsilon_\theta$ which is related to the score function as $s_\theta(x_t, t) = -\frac{-\epsilon_\theta(x_t, t)}{\sqrt{1-\bar{\alpha}_t}}$. We know the true score function for the given mixture of Gaussians, and we can estimate the learned score function using the model's output. In Figure 4, we plot the ground truth score (left) and the learned score (right) across various timesteps. We observe that the neural network learns a smooth approximation of the true score function, particularly around the regions between disjoint modes of the distribution from timesteps $t = 0$ to $t = 20$. Notice that the true score function has sharp jumps that separate two modes, however, the neural network can not learn such sharp functions and smoothly approximates a tempered version of the same. We also plot the estimated $\hat{x_0}$ and observe a smooth approximation of the step function instead of the exact step function. There is a region of uncertainty in the region between the two modes which leads to mode interpolation

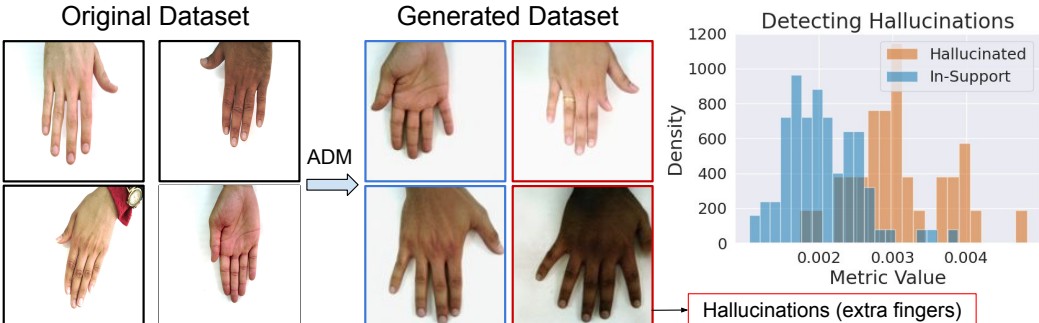

Figure 5: **Hands Dataset**. We train a ADM on the Hands dataset with 5000 images (first column) and show that the generated samples (second column) consists of hallucinated samples (additional/missing fingers). We then apply our proposed metric to detect these hallucinated samples (third column).

i.e sampling in the regions between the two modes. As another sanity check, we used the true score function in the reverse diffusion process for sampling (instead of the learned network). In this case, we did not see any instance of mode interpolation. This explains why the diffusion model generates samples between two modes of a Gaussian when it was never in the training distribution.

### 4.4 SIMPLE SHAPES

We now discuss the mode interpolation in the SIMPLE SHAPES dataset. In this context, the interpolation is not happening in the output space, but rather in the representation space. To investigate this, we performed a t-SNE visualization of the outputs from the bottleneck layer of the U-Net used in the Simple Shapes experiment, as shown in Figure 10. Regions 1 and 3 in the representation space semantically correspond to the images where squares are at the top and bottom of the image respectively. At inference time, we can see a clear emergence of region 2 which is between regions 1 and 3 (interpolated), and contains two squares (hallucinations) at the top and bottom of the image. This experiment concretely confirms that interpolation happens in representation space.

### 4.5 Mode Interpolation in Real World datasets: HANDS

We sought to demonstrate the occurrence of mode interpolation in a real-world setting. A well-documented challenge with popular text-to-image generative models is their difficulty in accurately generating human hands [28]. Despite extensive research in modern diffusion models, there is no conclusive explanation for the missing/additional fingers generated by these models. One hypothesis attributes this difficulty to the anatomical complexity of human hands, which involve numerous joints, fingers, and diverse poses. Another hypothesis suggests that, although large datasets contain many images of hands, these hands are often partially obscured (e.g., when a person is holding a cup) and occupy only a small region of the overall image.

To investigate this further, we trained a diffusion model on a datasets with high-quality images of human hands. The Hands dataset [1] consists of high resolution images of hands from 190 subjects of various ages. Each subject's right and left hands were photographed while opening and closing fingers against a uniform white background. We sample 5000 images from the Hands dataset and train an ADM [8] model on this dataset. We resize the images to 128x128 and use the same hyperparameters as that of the FFHQ dataset [18]. We mention all the hyperparameters in the Appendix A. We observe images with additional and missing fingers in the generated samples as seen in Figure 5. This is a pretty surprising result as it is non-trivial to assume that diffusion model generates images with additional fingers. Despite the potential for various failure modes, such as blurred hand images, these issues were not observed in our results. In some ways, the occurrence of 6-8 fingers is analogous to the occurrence of 2 squares in the SIMPLE SHAPES dataset. Thus, the presence of additional fingers in these images (i.e hallucinated images) generated by the diffusion model demonstrates the phenomenon of mode interpolation in real-world datasets. More example are shown in Fig. 20 & 21.

## 5 Diffusion Models know when they Hallucinate

Our previous sections established that hallucinations in diffusion models arise during sampling. More specifically, intermediate samples land in regions between different modes where the score

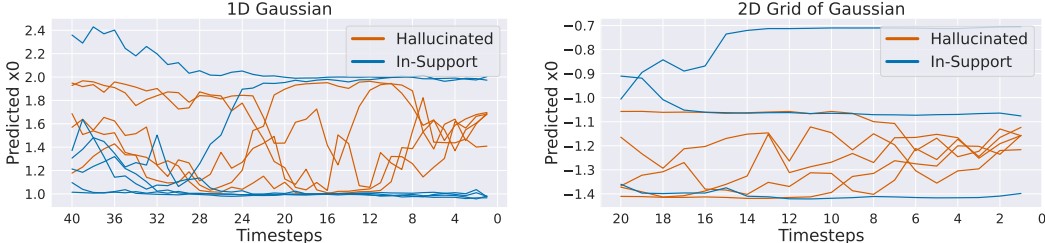

Figure 6: **Variance of $\hat{x}_0$ Trajectories**. The trajectory of the predicted $\hat{x}_0$ for hallucinated (shades of red), and non-hallucinated samples (shades of blue). We see that non-hallucinated samples stabilize in their prediction in the last 20 time steps for both 1D GAUSSIAN and 2D GAUSSIAN setups, whereas the hallucinated samples have high variance in the predicted $\hat{x}_0$ across time steps.

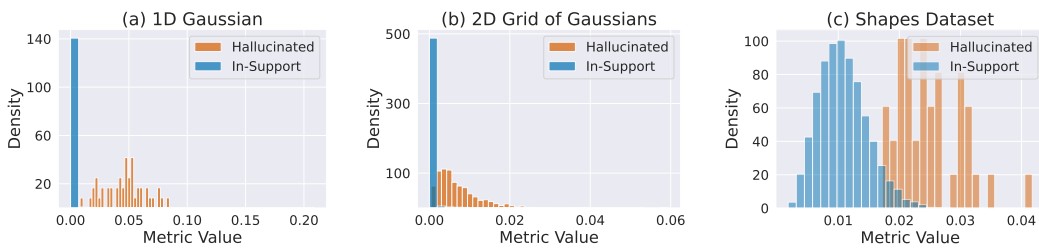

Figure 7: **Histogram of Hallucination Metric**. We depict the hallucination metric values for (a) 1D GAUSSIAN, (b) 2D GAUSSIAN, and (c) SIMPLE SHAPES setups. The histograms show that trajectory variance can capture a separation between hallucinated (orange) and non-hallucinated (blue) samples.

function has high uncertainty. Since neural networks find it hard to learn discrete 'jumps' between different modes (or a perfect step function), they end up interpolating between different modes of the distribution. This understanding suggests that the trajectory of the samples that generate hallucinations must have high variance due to the highly steep score function in the region of uncertainty. We will build upon this intuition to identify hallucinations in diffusion models.

## 5.1 Variance in the trajectory of prediction

We revisit the hallucinated samples in the 1D GAUSSIAN setup, and examine the trajectory of the predicted value of $\hat{x}_0$ during the reverse diffusion process. Figure 6 depicts the variance of trajectories leading to hallucinations (red shades) and those generating samples within the original data distribution (blue shades). For trajectories in shades of blue (non-hallucinations), the variance remains low beyond timestep $t = 20$. This indicates there is a minimal change in the predicted $\hat{x}_0$ during the final stages of reverse diffusion, signifying convergence. Conversely, the red trajectories (hallucinations) exhibit instability in the value of $\hat{x}_0$ in the same region. This suggests a high overall variance in these trajectories.

## 5.2 Metric for detecting hallucination

Based on the above observation about high variance in predicted values of $x_0$ in the reverse diffusion process, we use the same observation as a metric to distinguish hallucinated and non-hallucinated (in-support) samples. The intuition behind the metric is to capture the variance in the trajectory of $\hat{x}_0$. Let $T_1$ be the starting timestep and $T_2$ be the end timestep. Mathematically, the metric can be defined as follows:

$$\texttt{Hal}(x) = \frac{1}{|T_2 - T_1|} \sum_{i=T_1}^{T_2} \left( \hat{x_0}^{(i)} - \overline{\hat{x_0}^{(t)}} \right)^2 \tag{4}$$

where $\hat{x_0}^{(t)}$ represents the predicted values of the final image at different time steps $(t)$, and $\overline{\hat{x_0}^{(t)}}$ is the mean of these predictions over the same time steps. We now utilize this metric to analyze the histogram values of each sample from the three experimental setups studied thus far. This metric can be implemented in two ways. One approach is to store $\hat{x_0}$ during the reverse diffusion process and then compute the variance. Alternatively, we explore a method where forward diffusion is performed for $k$ steps between $T_1$ and $T_2$, predicting $\hat{x_0}$ at each step, and then computing the variance.

**SIMPLE SHAPES.** In the SIMPLE SHAPES setup, a sample is labeled as hallucinated if more than one shape of the same type occurs in the generated image. We generate 7500 images using a DDPM and study the separation between hallucinated and non-hallucinated images. We find that the reverse diffusion process of $T = 1000$ steps is rather long. Generally, the image stabilizes around $T = 700$ (as shown in Appendix 18). Therefore, we use the time range between $T = 850$ and $T = 700$ in the reverse diffusion process to compute the variance of the predicted sample value. Using this process, we can filter out 95% of the hallucinated samples while retaining 95% of the in-support samples. The histogram for the values is presented in Figure 7.

**1D GAUSSIAN.** In the 1D-Gaussian setup, we label any examples as a hallucination if they have negligible probability (for instance values greater than $6\sigma$ from the mean under normal) under the real data distribution (refer to Figure 2). We measure the variance of the last 15 steps of the $\hat{x_0}$ during the reverse diffusion process, and plot the histogram of values of the same in Figure 7. We can filter out 95 % of the hallucinated samples while retaining 98% of the in-support samples.

**2D GAUSSIAN.** Next, we discuss our investigation on synthetic datasets with experiments on the 2D GAUSSIAN dataset. Similar to the 1D GAUSSIAN setup, we once again measure the prediction variance of the last 20 steps of the reverse diffusion process. We compute the variance per dimension and then take the mean across dimensions to . With this metric, we can filter out 96% of the hallucinated samples while retaining 95% of the in-support samples.

**HANDS.** Finally, we conclude our investigation with experiments on the Hands dataset. To analyze the effectiveness of the proposed metric, we manually label 130 images from the generated samples as hallucinated vs. in-support. This includes 88 images with 5 fingers and 40 images with missing/ additional fingers i.e. hallucinated samples. The histogram (in Figure 5) shows that the proposed metric can indeed detect these hallucinations to a reasonable degree. In our experiments, we observe that we can eliminate 80% of the hallucinated samples while retaining 81% of the in-support samples. The trajectories of the hallucinated and in-support samples are shown in Figures 22 and 23, respectively. A higher variance in the trajectory of $\hat{x_0}$ is clearly observed in the hallucinated samples compared to the in-support samples. We note that the detection is a hard problem and the fact that the method transfers to the real world is proof of the relationship between mode interpolation and hallucination in real-world data.

## 6    Implications on Recursive Model Training

The internet is increasingly populated by more and more synthetic data (data synthesized from generative models). It is likely that future generative models will be exposed to large volumes of machine-generated data during their training [26, 27]. Recursive training on synthetic data leads to mode collapse [2, 9] and exacerbates data biases. In this section, we study the impact of hallucinations within the context of recursive generative model training. We adopt the standard synthetic-only setup similar to [2] where we only use synthetic data from the current generative model in training the next generation of generative models. The first generation of generative model is trained on real data and samples from this generative model is used to train the second generation (and so on).

Most of the previous works [3] studied the model collapse to a single mode. In this work, we emphasize that the interaction between modes and mode interpolation plays a massive role when training generative models on their own output.

**2D GAUSSIAN.** When we recursively train a DDPM on its own generated data using a square grid of 2D Gaussians (with $T = 500$), the hallucinated samples significantly influence the learning of the next generation's distribution (see Figure 9). The frequency of the interpolated samples increases as we further train on the learned distribution that consists of interpolated samples. Figure 9d shows samples from Generation 20, where it is evident that the modes have almost collapsed into a single mode, differing greatly from the original data distribution.

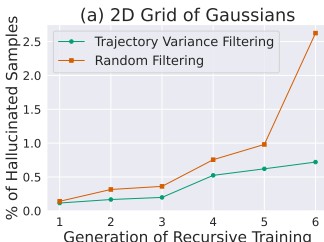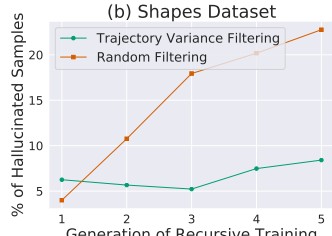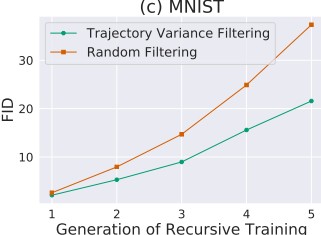

Figure 8: **Mitigating Hallucinations with Pre-emptive Detection**. We filter out hallucinated samples using the metric from § 5 before training on samples from the previous generation of the diffusion model. In the case of (a) 2D GAUSSIAN, (b) SIMPLE SHAPES, where we have clear definitions of hallucination (mode interpolation, and new shape combinations) we see the effectiveness of our variance-based filtering method in minimizing hallucinations across generations compared to random filtering. In the case of (c) MNIST dataset, we measure the FID of subsequent generations and notice that pre-emptive filtering of hallucinated samples makes the recursive model collapse slower.

**SIMPLE SHAPES.** We define a hallucinated sample as one that contains at least two shapes of the same type (which is never seen in the training distribution). We observe the presence of around 5% hallucinated samples when trained on the real data. We note that the ratio of hallucinated samples increases exponentially as the we iteratively train the diffusion model on its own data. This is expected as the diffusion model progressively learns from a distribution increasingly dominated by hallucinated images, compounding the effect in subsequent generations.

**MNIST.** We also run the recursive model training on the MNIST dataset [22]. At every generation, we generate 65k images and sample 60k images using the filtering mechanism. For each generation, we train a class conditional DDPM with Classifier-Free Guidance [16] with $T = 500$ for 50 epochs. To evaluate the quality of the generated images, we compute the FID [14] using a LeNet [22] trained on MNIST instead of Inception backbone as MNIST is not a natural image dataset. In Figure 8, we clearly see that the proposed metric based on the variance of the trajectory outperforms the random filtering method across all generations (lower FID is better). We also plot the Precision and Recall [36] curves (in the Appendix Figure 18) where we observe that our filtering mechanism selects high quality samples without much loss in diversity.

**Mitigating the curse of recursion with pre-emptive detection of hallucinations.** Based on the metric developed in § 5, we analyze the efficacy of the proposed metric in filtering out the hallucinated samples for the next generation of training. After training each generation of the generative model, we sample $k$ images more than size of the training data and then filter out hallucinated samples based on the metric. Figure 8 shows the results on 2D Grid of Gaussians, SIMPLE SHAPES and MNIST dataset. We also compare with random filtering where we randomly sample points for the next generation. The variance-based filtering method easily outperforms the random sampling method in all the generations. We see the effectiveness of the proposed metric in minimizing the rate of hallucinations across generations and thus model collapse to a certain extent. This holds true for all the three datasets we have studied in this work.

## 7 Discussion

In this work, we performed an in-depth study to formulate and understand hallucination in diffusion models, focusing on the phenomenon of mode interpolation. We analyzed this phenomenon in four different settings: 1D Gaussian, 2D Grid of Gaussians, Shapes and Hands datasets, and saw how diffusion models learn smoothed approximations of disjoint score functions, leading to mode interpolation. Based on our analysis, we developed a metric to identify hallucinated samples effectively and explored the implications of hallucination in the context of recursive generative model training. This study is the first to propose mode interpolation as a potential hypothesis for explaining the generation of additional fingers in large-scale generative models. We hope that future research will build upon this hypothesis and develop methods to mitigate these issues in generative models. We hope our work inspires future research in understanding and mitigating hallucination in diffusion models.

## Acknowledgements

PM is supported by funding from the DARPA GARD program. ZL gratefully acknowledges the NSF (FAI 2040929 and IIS2211955), UPMC, Highmark Health, Abridge, Ford Research, Mozilla, the PwC Center, Amazon AI, JP Morgan Chase, the Block Center, the Center for Machine Learning and Health, and the CMU Software Engineering Institute (SEI) via Department of Defense contract FA8702-15-D-0002, for their generous support of ACMI Lab's research. ZK gratefully acknowledges support from the Bosch Center for Artificial Intelligence to support work in his lab as a whole.

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

# A Additional Experimental Details

## A.1 Gaussian experiments

We run all our experiments for $10,000$ epochs with batch size of $10,000$. A linear noise schedule is used with starting noise $\beta_0 = 0.001$ and the final noise $\beta_1 = 0.2$. We use $T = 1000$ by default in our experiments (unless specified otherwise). The neural network (NN) is trained to predict the noise (similar to the original DDPM [15] implementation) and we use a Mean Squared Error loss to train the model. The input and output of the NN have the same shape (in this case, 1 for 1D Gaussian and 2 for the 2D Gaussian). The NN architecture starts with an initial fully connected layer, followed by three blocks and then output fully connected layer. Each block includes normalization, a LeakyReLU activation, and two fully connected layers. Finally, the output is normalized and transformed back to the input dimension with a fully connected layer. Adam [21] with learning rate of 0.001 is used as the optimizer. We build our codebase on top of [1] for the synthetic toy experiments.

**Metric**: We use $t = 0$ to $t = 15$ (last 15 steps in the reverse diffusion process) to compute the variance of the trajectory in the case of Gaussian 1D and $t = 0$ to $t = 8$ in the case of 2D Gaussian Grid.

## A.2 Shapes

The generated images are grayscale images of size $64 \times 64$. A total of 5000 images is generated for training the diffusion model. We use a U-Net [34] architecture to model the reverse diffusion process. We use a cosine noise scheduler similar to ADM [29]. We derive our implementation based on [2] for training the DDPM. We train an unconditonal DDPM on the dataset with $T = 1000$ while training and 250 steps during sampling to reduce computation cost [40].

## A.3 MNIST

MNIST [22] consists of 60,000 grayscale images of size (28, 28). We use classifier-free guidance [16] to train a conditional DDPM on MNIST with $T = 500$. For each generation, we train for a total of 50 epochs with a batch size of 512 shared across 4 GPUs. Adam [21] optimizer with learning rate of 1e-4 is used to train the network. We use a U-Net [34] with 256 feature dimension to model the reverse diffusion process. For the variance filtering mechanism in Section 6, we use 10 timesteps between $t = 100$ to $t = 150$ to compute the variance of the trajectory. In the case of MNIST, we do post-hoc filtering just using the samples. This means that we add $t$ timesteps of noise, then compute $\hat{x_0}$ and then use this to compute variance.

Our implementations of the DDPM model is based on PyTorch [31].

**Compute**: We run all our experiments of Nvidia RTX 2080 Ti and Nvidia A6000 GPUs. The training and sampling for the Gaussian experiments takes less than 3 hours on single 2080Ti GPU. Sampling 100 million datapoints takes around 3-4 hours. Running DDPM on the shapes dataset takes around 6-7 hours with 4 2080Ti GPUs. The recursive generative training on MNIST takes about 16 hours with 4 A6000 GPUs for 5 generations.

## A.4 HANDS

ADM refers to Ablated Diffusion model as defined in [8]. We trained for a total of 200k iterations with batch size 16 and a learning rate of 1e-4. The diffusion process was trained with 1000 timesteps ($T = 1000$) with a cosine noise schedule. The U-Net comprised 256 channels, with an attention mechanism incorporating 64 channels per head and 3 residual blocks. For sampling, we use 500 timesteps with respacing. We base our implementation and hyperparameters on the official DDPM-IP [30] repository.

---

[1] https://github.com/tqch/ddpm-torch
[2] https://github.com/VSehwag/minimal-diffusion

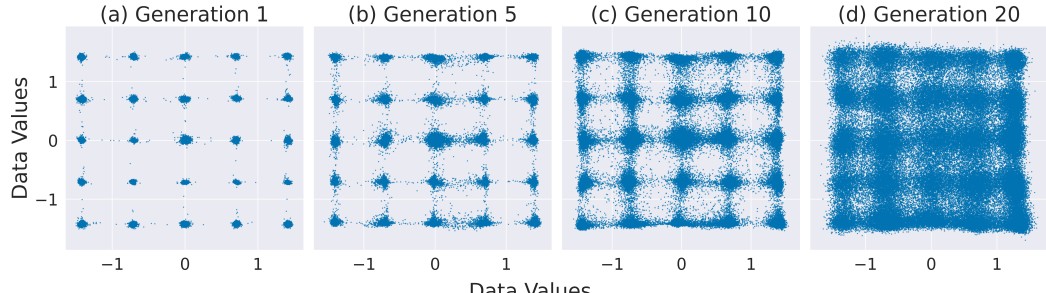

Figure 9: **Recursive Training on 2D GAUSSIAN**. We investigate the impact of recursively training a DDPM on its own generated data using a square grid of 2D Gaussians with $T = 500$ diffusion steps. In each generation, we sample 100k examples, and train the subsequent generation on these data points. As the training progresses through multiple generations, the hallucinated (interpolated) samples significantly influence the learning of the next generation's distribution.

## B Limitations and Broader Impact

Hallucinations in LLMs have been studied extensively [46, 47] given the widespread use of these systems in various contexts. This work investigates hallucinations in diffusion models. In current generative models, these hallucinations could be used to more easily identify machine-generated images. Developing a metric to identify these hallucinations and remove them could make the detection of generated images much harder. However, we argue that understanding hallucinations in diffusion models is crucial as it can help shed light on their failure modes and thereby enable better control in practical applications.

In current text-to-image generative models, the poorly modeled "hands" are a clear giveaway in identification of AI generated images. The detection of such AI-generated content would be made much more difficult if these hallucinations were identified and removed from the generated images. While our work builds an understanding of hallucinations, and allows us to also detect them, we believe that future generations of models would have become more robust to such hallucinations by virtue of training on more data independent of this work.

Concerning the limitations of the proposed hallucination metric, the selection of the right timesteps is key to be able to detect hallucinations. More analysis on what region of trajectory leads to hallucinations would be useful across various schedules and sampling algorithms. We believe these are great areas for future work to explore. Additional explorations of mode interpolation and hallucinations in real-world datasets would be useful to the community.

## C Additional Experiments and Figures

We also study Variational Diffusion Models (VDM) [20] to verify the generality of our findings. Our results show that the over-smoothed score function phenomenon persists in VDM, supporting the hypothesis that this issue is not specific to DDPM. We train a simple VDM on the 2D Gaussian with 10k samples. We follow the setup and hyperparameters in the official implementation [3]. We train both continuous and discrete variants of VDM on the 2D Gaussian dataset. The main observation is that VDM mitigates the hallucinations significantly especially with more training data but the phenomenon of mode interpolation still exists. In this figure, we also show the impact of the number of sampling steps on the count of hallucinations. We clearly see that increasing the number of sampling steps reduces the number of hallucinated samples. This can be clearly observed in Figure 11 (first two columns) where the count of hallucinations decreases mode interpolation.

The frequency of mode interpolation is inversely proportional to the number of training samples. We train the unconditional diffusion model with 25k, 50k, 100k and 500k samples from the true distribution.

---

[3]https://github.com/google-research/vdm

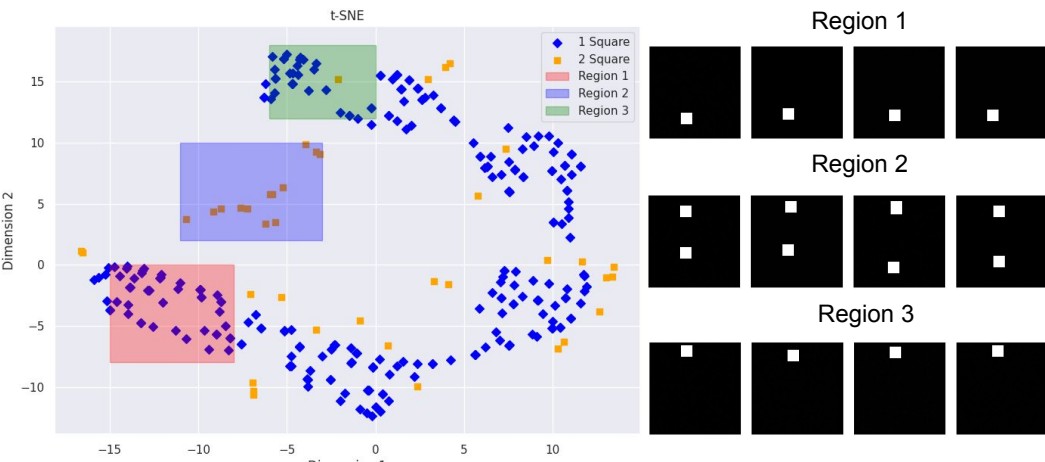

Figure 10: **Interpolation in Representation Space**. We analyze the bottleneck of the U-Net to demonstrate mode interpolation in the Shapes dataset. We clearly see that Region 2 (which consists of 2 squares) is interpolating between Region 1 (one square in the bottom half) and Region 3 (one square in the top half).

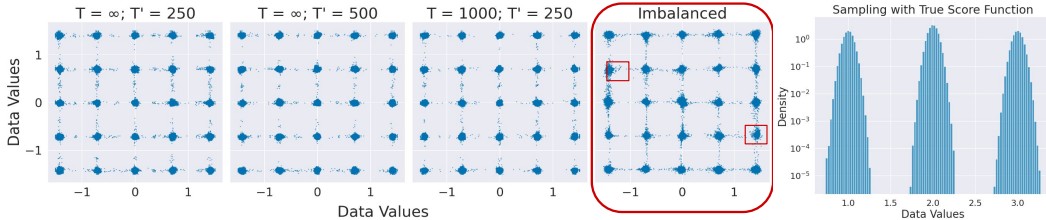

Figure 11: **Variational Diffusion Model**. We train a Variational Diffusion Model (VDM) on the 2D Gaussian Data with 10k samples (first three columns). $T$ denotes the timesteps during training and $T'$ denotes the sampling timesteps. $T = \infty$ refers to the continuous time variant. The fourth column shows a DDPM trained on a 2D Gaussian with imbalanced modes. The boxes indicate the modes with less data. The last column shows result of sampling from the true score function.

1. Figure 12 shows the histogram of samples generated by the diffusion model (with 10 million samples) when the model is trained on the distribution with $\mu_1 = 1, \mu_2 = 2, \mu_3 = 3$.

2. Figure 13 shows the histogram of samples generated by the diffusion model (with 10 million samples) when the model is trained on the distribution with $\mu_1 = 1, \mu_2 = 2, \mu_3 = 4$.

3. We also experiment with mixture of 2 Gaussians in Figure 14 and 4 Gaussians in Figure 15.

4. Figure 16 shows the FID, precision and recall curves for MNIST across generations.

5. Figure 17 shows additional examples of hallucinated images generated by the diffusion model.

6. Figure 18 shows the $\hat{x}_0$ across various timesteps for a hallucinated image. The number on top of the image indicates the timestep.

7. Figure 19 shows the $\hat{x}_0$ across various timesteps for a image in-support of the distribution. The number on top of the image indicates the timestep.

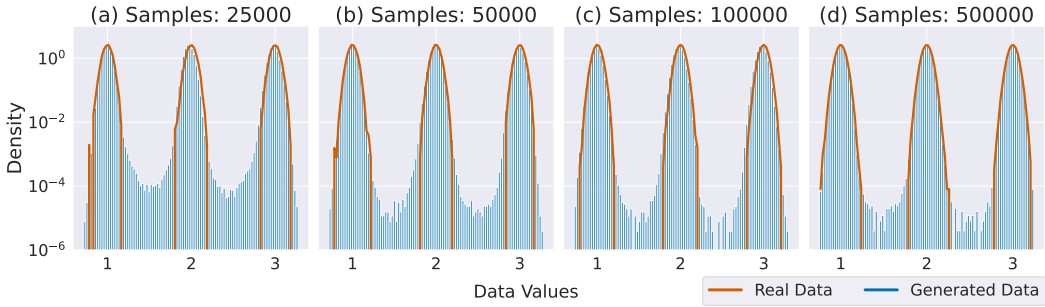

Figure 12: Mixture of 3 Gaussians with $\mu = [1, 2, 3]$. We vary the number of training samples and observe that mode interpolation decreases with increase in the size of training data

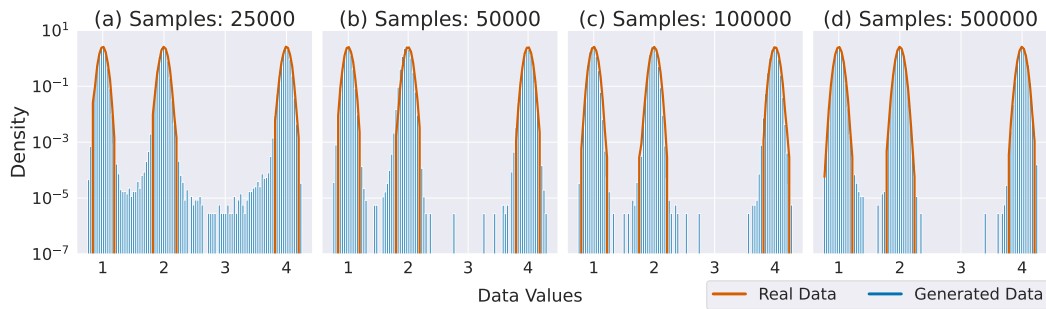

Figure 13: Mixture of 3 Gaussians with $\mu = [1, 2, 4]$. We vary the number of training samples and observe that mode interpolation decreases with increase in the size of training data.

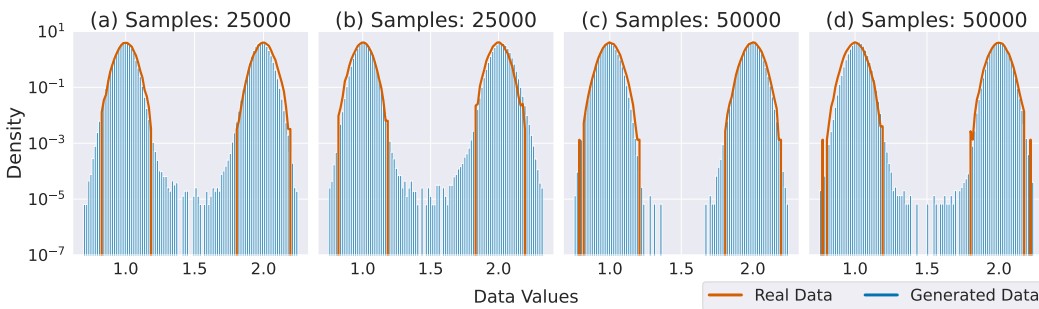

Figure 14: Mixture of 2 1D Gaussians with varying number of training samples. (a) and (b) have the same number of training samples but with two different seeds. Similarly for (c) and (d).

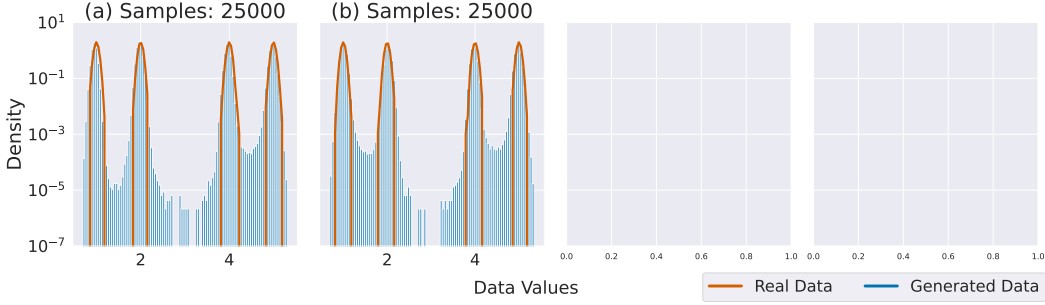

Figure 15: Mixture of 4 1D Gaussians ($\mu = [1, 2, 4, 5]$) with varying number of training samples. (a) and (b) have the same number of training samples but with two different seeds. We clearly see more samples in the region between modes $\mu_1 = 1$ and $\mu_2 = 2$ compared to $\mu_2 = 2$ and $\mu_3 = 4$.

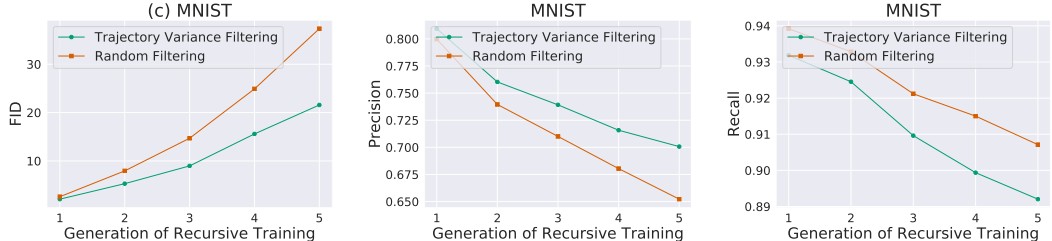

Figure 16: Recursive Generative Training on MNIST with Variance and Random Filtering. We observe that the proposed filtering mechanism can discard low quality samples while maintaining sufficient diversity.

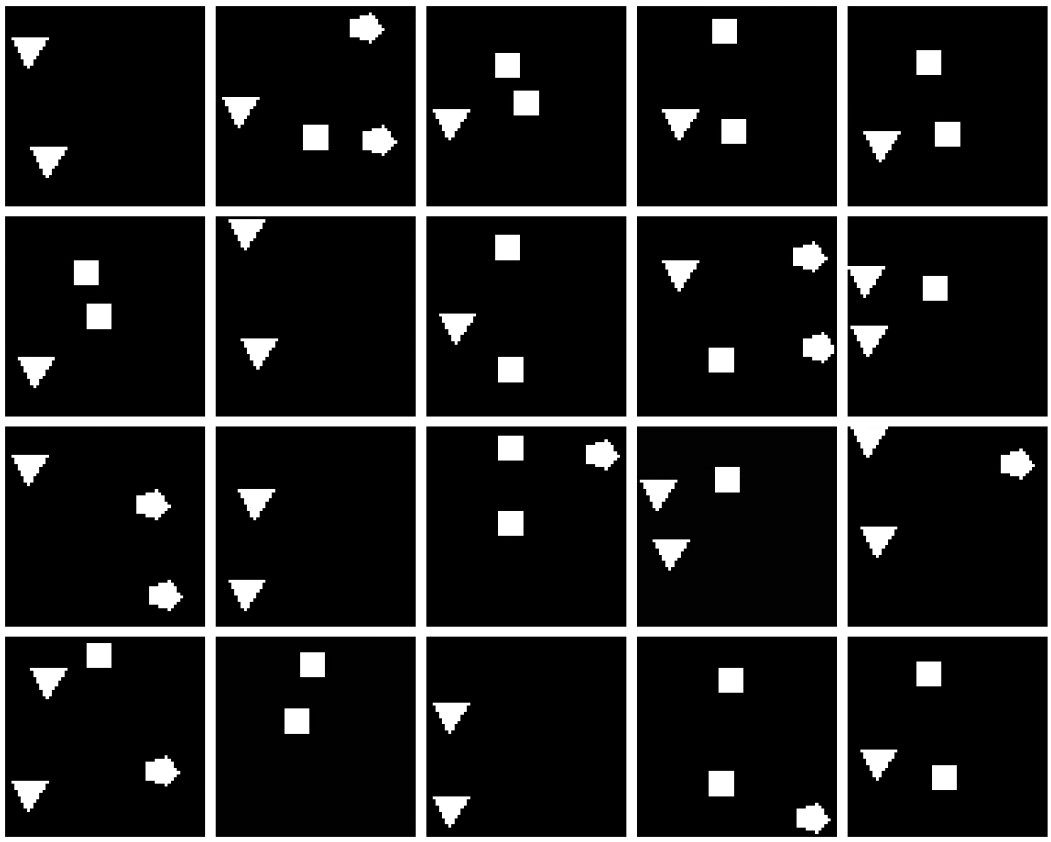

Figure 17: Example of Generated Hallucinated Images

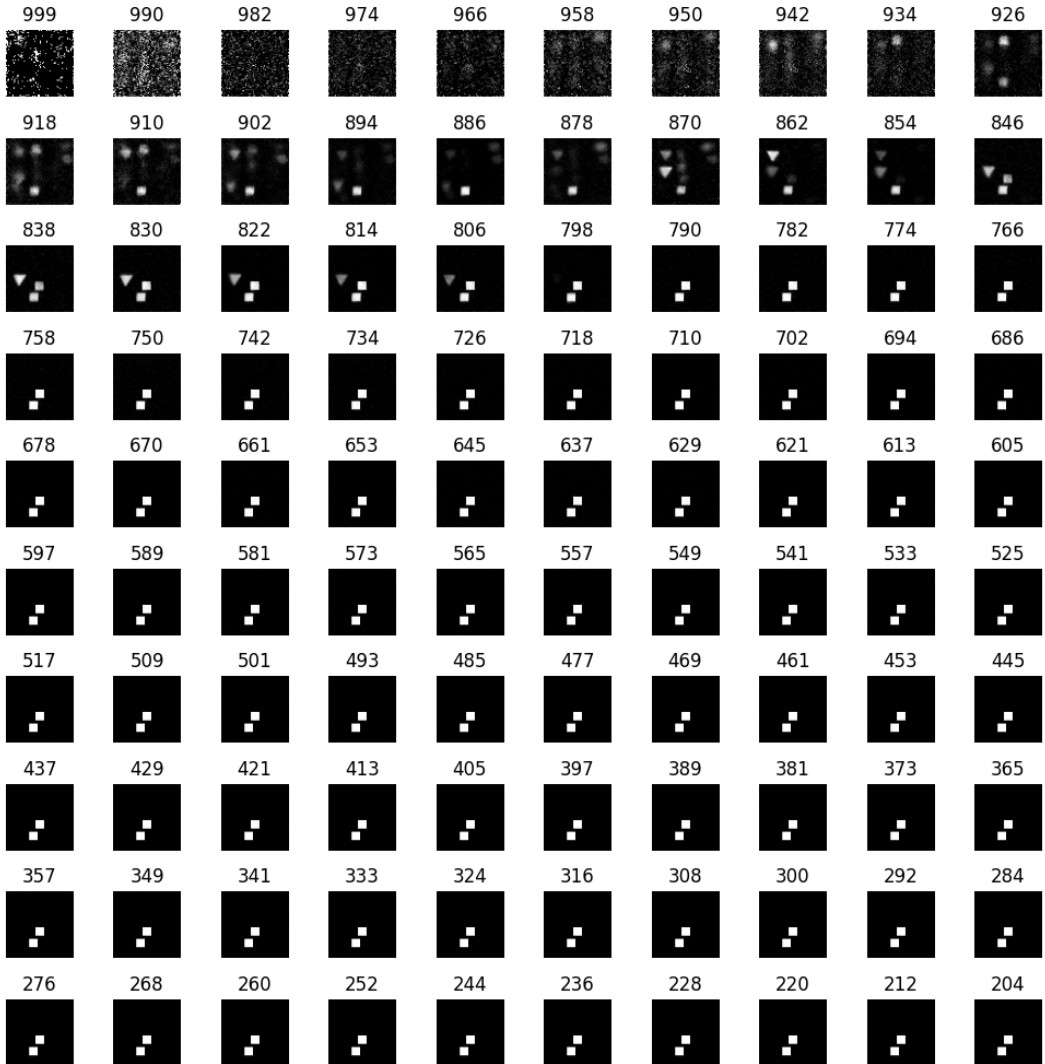

Figure 18: $\hat{x}_0$ for Hallucinated Sample. Here, we observe that the predicted $x_0$ has a lot of variance around $t = 700$ to $t = 850$. This clearly motivates our proposed metric.

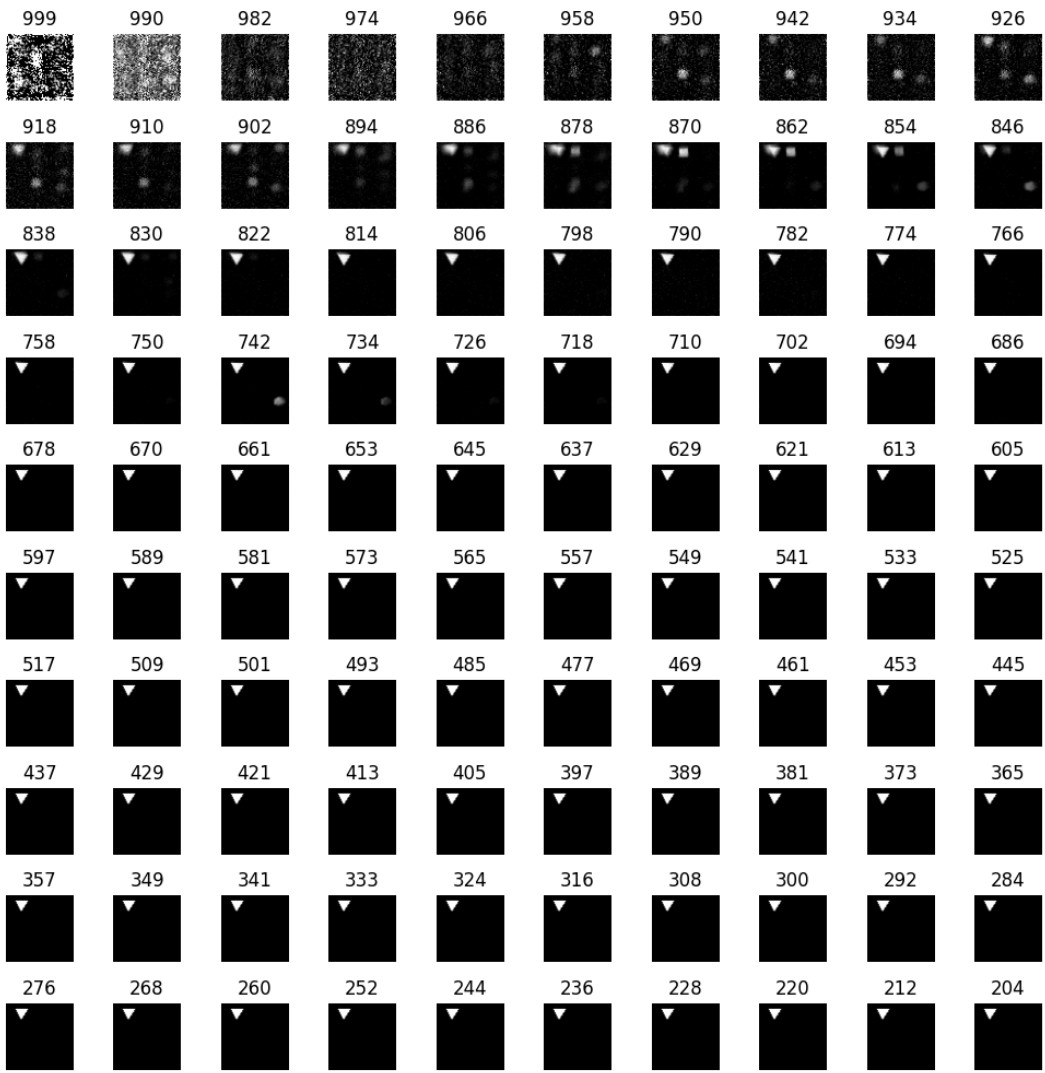

Figure 19: $\hat{x}_0$ for In-Support Sample. Here, we observe that the predicted $x_0$ is more consistent around $t = 700$ to $t = 850$.

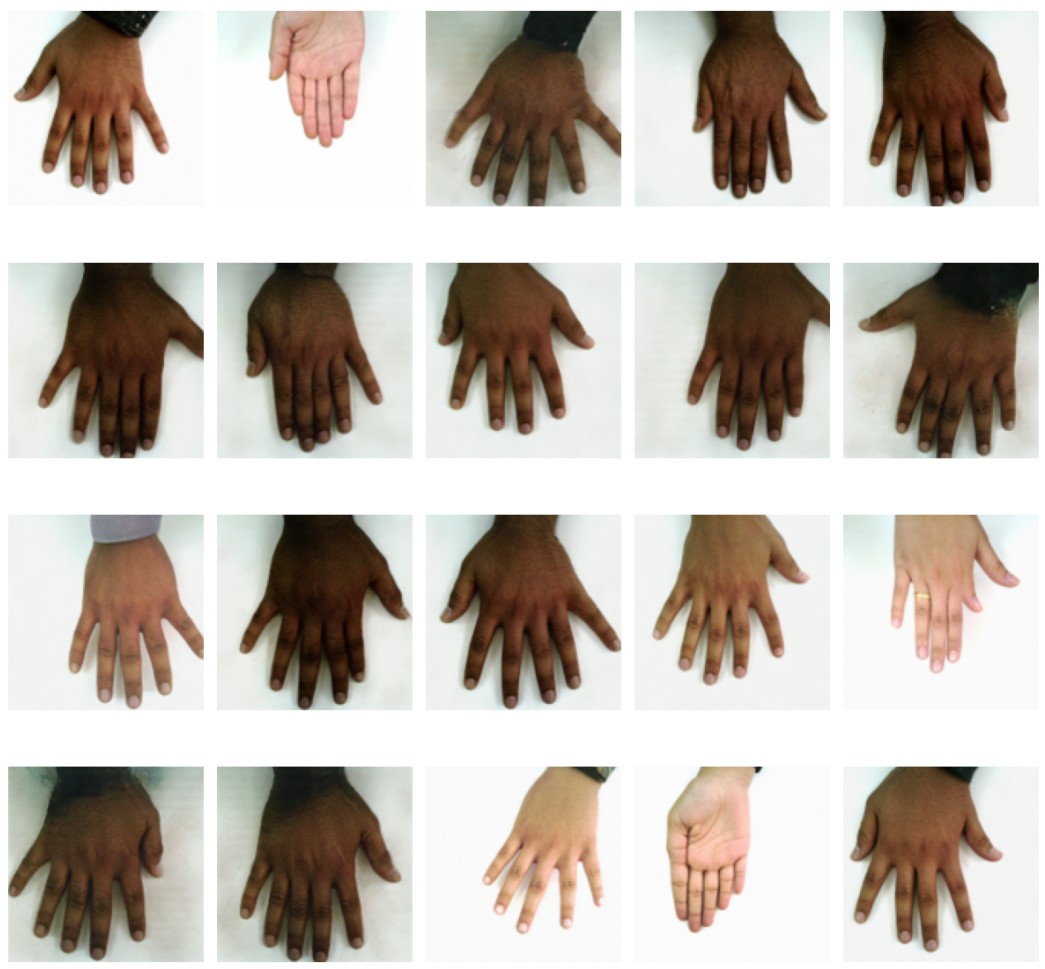

Figure 20: Hallucinated Images of Hands generated by the diffusion model.

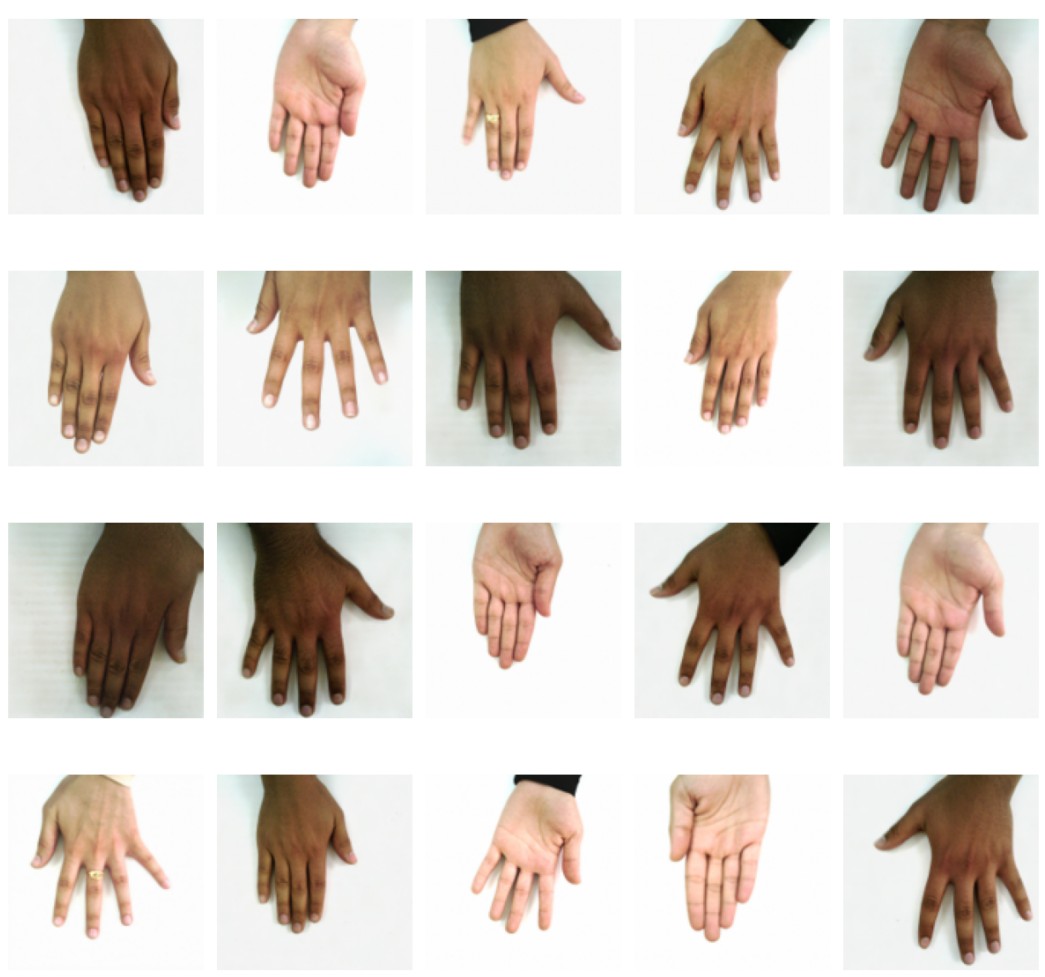

Figure 21: In-Support Images of Hands generated by the diffusion model.

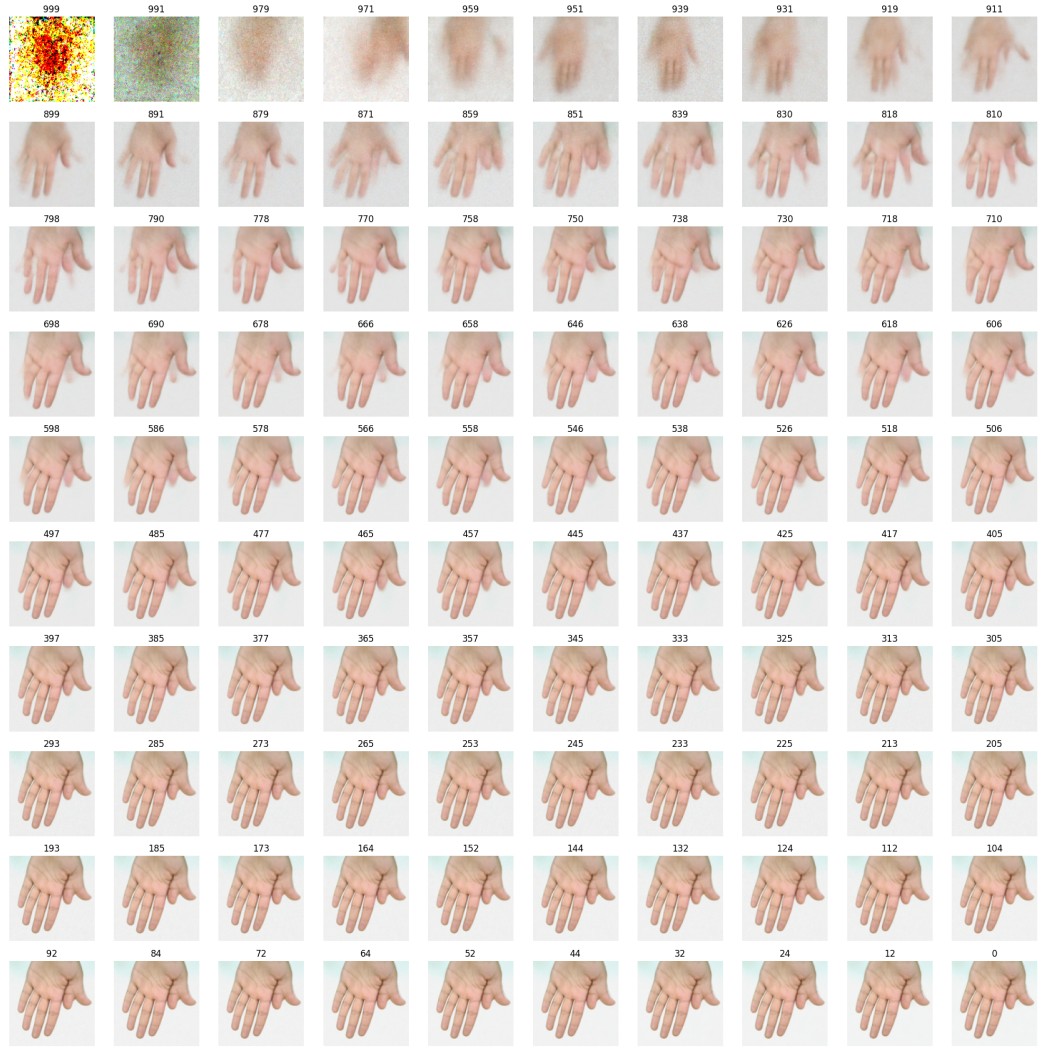

Figure 22: $\hat{x}_0$ trajectory for Hallucinated Sample (with 250 timesteps). We observe high variance/instability during the steps $t = 600$ to $t = 900$.

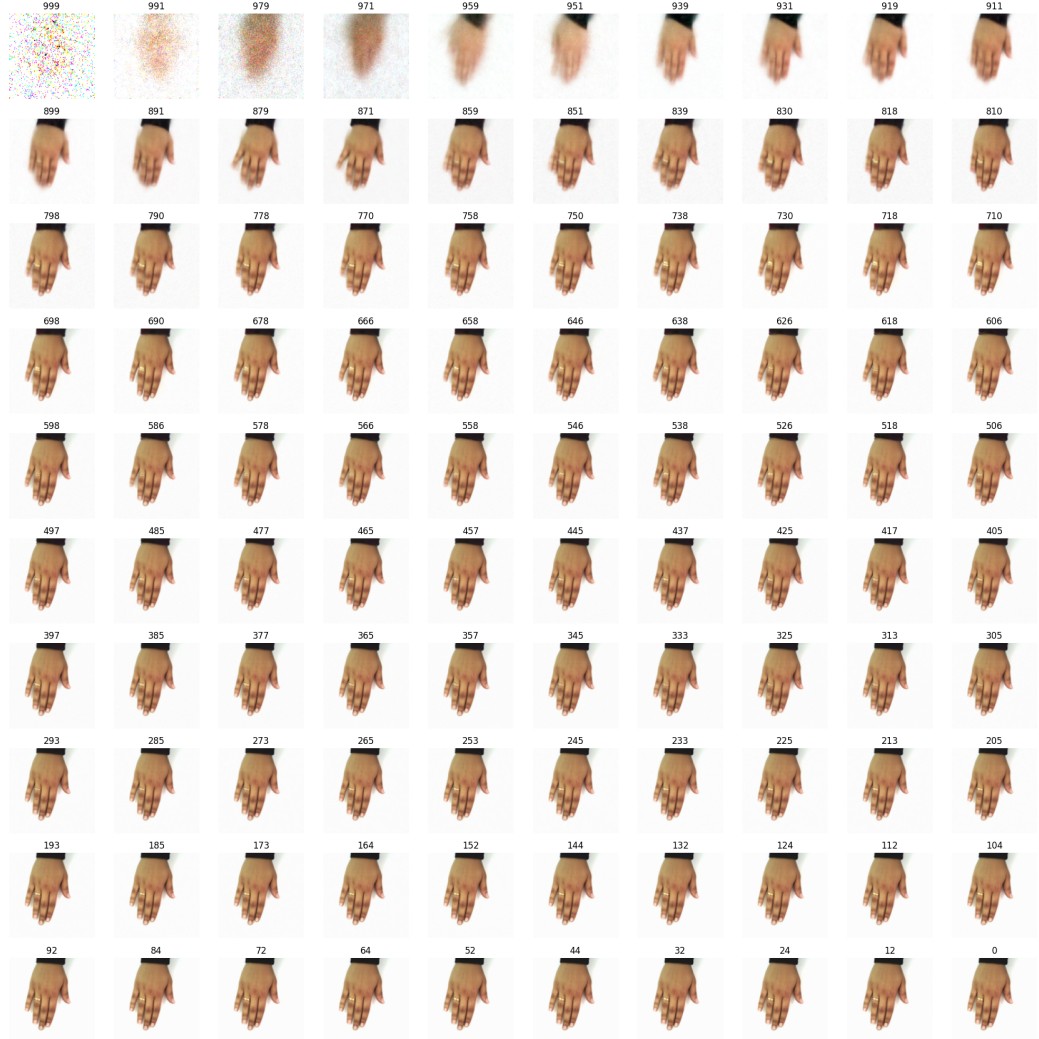

Figure 23: $\hat{x}_0$ trajectory for In-Support Sample (with 250 timesteps). We do not observe high variance/instability during the steps $t = 600$ to $t = 900$.

