# OpenReview forum: "Understanding Hallucinations in Diffusion Models through Mode Interpolation"
_NeurIPS.cc/2024/Conference — NeurIPS 2024 poster_

### Official Review · Reviewer_yx5u · 2024-07-07

**Soundness:** 3
**Presentation:** 4
**Contribution:** 3
**Rating:** 5
**Confidence:** 3

**Summary:**

The paper introduce the concept of hallucinations for image diffusion models.

The key contributions includes:

1. Definition of Hallucinations: Hallucinations are defined as samples generated by diffusion models that lie completely outside the support of the real data distribution.

2. Mode Interpolation Phenomenon: The phenomenon where diffusion models interpolate between nearby data modes, generating samples that do not exist in the original training data. This smooth interpolation leads to artifacts, termed hallucinations​​.

3. Causes of Hallucinations: Hallucinations are attributed to the smooth approximation of discontinuous loss landscapes by the diffusion model. This leads to interpolation between distinct data modes that are not present in the original dataset​​.

4. Experimental Findings: Experiments with 1D and 2D Gaussians show that hallucinations occur due to mode interpolation, particularly between nearby modes. The variance in the trajectory of the generated sample increases towards the end of the backward sampling process, indicating out-of-support samples​​.

5. Mitigation of Hallucinations: A simple metric to capture the high variance in the sample trajectory can effectively remove over 95% of hallucinations during generation while retaining 96% of in-support samples​​.

6. Impact on Recursive Training: The removal of hallucinations has implications for the collapse and stabilization of recursive training on synthetic data. Experiments on datasets like MNIST and 2D Gaussians demonstrate the effects​​.

**Strengths:**

1. The paper introduces Hallucinations in diffusion models, which is an under-explored area. The paper's focus on mode interpolation as a source of these hallucinations brings a fresh perspective.
2. The paper exhibits high-quality research through its comprehensive experimental design and thorough analysis.
3. The paper is well-written and clearly structured, making it accessible to both experts and those new to the field.
4. The significance of the paper lies in its potential impact on the development and refinement of diffusion models. By identifying and addressing the issue of hallucinations, the research provides valuable insights that could lead to more reliable and accurate generative models.

**Weaknesses:**

1. SMLD [1] claims that `the scarcity of data in low density regions can cause difficulties for both score estimation with score
matching and MCMC sampling with Langevin dynamics.` SMLD [1] addresses this problem by introducing slow mixing of Langevin dynamics. The datasets used in the paper have the same density for different modes. I think the hallucination phenomenon with different density for different modes should also been explored.
2. While the paper provides a robust analysis using 1D and 2D Gaussian datasets, its experimental scope is somewhat limited. These simplified datasets may not fully capture the complexity of real-world data distributions. Is the hallucination of diffusion models on real image distribution (such as face) helpful to its diversity due to mode interpolation?
3. While the paper touches on the implications of hallucinations for recursive training stability, this discussion is relatively brief and lacks depth. Given the potential significance of this aspect, a more extensive exploration of how hallucination mitigation affects recursive training dynamics would have been valuable. This could include detailed experiments and analyses on the long-term effects of hallucination removal on model performance and stability.

[1] Generative Modeling by Estimating Gradients of the Data Distribution

**Questions:**

Please see weaknesses.

**Limitations:**

The authors have adequately described the limitations and potential negative societal impact of their work.

---

> ### Author Rebuttal · Authors · 2024-08-07
>
> We are happy to see that you liked the fresh perspective on hallucinations in diffusion models and mode interpolation presented in our work, and found the paper to showcase high-quality research, and comprehensive experimental design, and have a potential impact on the development of more reliable generative models. We acknowledge your concerns and attempt to respond to them line by line below:
>
> ### **Re: Testing on Natural Images**
>
> Please refer to the [global response here](https://openreview.net/forum?id=aNTnHBkw4T&noteId=2W1mcxDdVO) along with the figures in the **attached PDF** for an exciting update with results on the Hands-11k dataset!
>
>
> ### **Re: Hallucinations with Imbalanced Density Distribution across Modes**
>
> We conducted additional experiments using datasets with varying densities for different modes. Specifically, we trained a DDPM on the Gaussian 2D dataset with two modes that have only 1/100th of the samples when compared to the other modes. In the **attached PDF**, see the modes highlighted with a red square. We observed that imbalanced data can exacerbate mode interpolation near these underrepresented modes. These experiments demonstrate that the hallucination phenomenon persists even with differing densities, further validating our hypothesis.
>
> ### **Is the hallucination of diffusion models on real image distribution (such as face) helpful to its diversity due to mode interpolation?**
>
> Hallucinations are one of the overlooked failure modes in diffusion models. Certain hallucinations can indeed introduce novel and creative variations that may not be present in the training data. However, in this work, we do study hallucinations in a concrete setup of unconditional diffusion models, where samples emerge at generation in otherwise zero-density regions in the real data manifold. For instance, you may look at Figure 2 in the **attached PDF**. We believe images in such zero-density regions manifest in unwanted characteristics such as incorrect hands. Though this may indeed be a double-edge sword. We are in a way stopping the model from going out of the data manifold, which can be useful for abstract/creative tasks where hallucinations may be desirable.
>
> ###  **Long-Term Effects on Recursive Training**:
>
> We discuss the long-term effects of hallucinations in recursive model training in Section 6. We show that filtering hallucinated samples can mitigate model collapse in this setting (Figure 8). We also study the long-term impact of hallucinations in Gaussian 2D (Figure 7).
>
> A practically relevant setting is the data accumulation setting as discussed in [1]. The key difference is that in their setting synthetic data is accumulated along with real data as the training progresses across generations. In Section 6, we only use the synthetic data sampled from the most recent generative model. Gerstgrasser et. al argues that model collapse can be avoided with their data accumulation pipeline. However, we argue that mode interpolation can provide a novel viewpoint in this framework as the subsequent generations would result in a much higher fraction of hallucinations, *even if* real data is included in subsequent generations. We will dedicate an extended portion of our paper to discuss the important implication and agree with your characterization of its importance.
>
> [1] Gerstgrasser, Matthias, et al. "Is model collapse inevitable? breaking the curse of recursion by accumulating real and synthetic data." arXiv preprint arXiv:2404.01413 (2024).
>
> ---
>
> Please refer to the **attached PDF** for detailed results and figures from our additional experiments. We hope we are able to further strengthen your conviction and support for our work through this rebuttal. Please let us know if you have any remaining concerns.

---

> > ### Comment · Reviewer_yx5u · 2024-08-11
> >
> > I acknowledge having read the authors' rebuttal. My overall assessment of the paper remains unchanged, and I continue to support my current rating.

---

### Official Review · Reviewer_Zx7y · 2024-07-09

**Soundness:** 2
**Presentation:** 2
**Contribution:** 2
**Rating:** 3
**Confidence:** 5

**Summary:**

This paper addresses the *hallucination* phenomenon in diffusion models, where generated samples fall outside the training distribution. The authors propose *mode interpolation* as an explanation. They analyze synthetic datasets and conclude that hallucinations occur between nearby modes due to the inability of deep networks to learn ground truth score functions with sharp jumps for small timesteps $t$. The authors introduce a metric along the denoising trajectory to identify hallucinated samples and demonstrate that filtering out these samples during recursive training improves the quality of generated samples on both synthetic and real-world datasets.

**Strengths:**

1. The paper explores a novel phenomenon of hallucinations in diffusion models and provides a plausible explanation through mode interpolation.
2. The authors employ sound toy experiments to validate the hallucination phenomenon and their proposed explanation.

**Weaknesses:**

1. **Mode interpolation as an explanation for hallucinations is not entirely convincing.**
   The concept of mode interpolation is proposed and validated through experiments on synthetic Gaussian mixture datasets. However, the relationship between mode interpolation and hallucinations in more complex datasets and models (such as extra or missing fingers in StableDiffusion) remains unclear, particularly for latent diffusions with decoders. The authors should provide more evidence of mode interpolation in complex cases and hallucinated samples. Furthermore, the role of the decoder in hallucinations, mentioned in the abstract, is not adequately addressed in the discussion.

2. **Experiments should be more comprehensive and robust.**
   - *Comprehensive experiments:* The authors primarily use synthetic Gaussian mixture and simple shapes datasets to validate mode interpolation and hallucination removal. Additional experiments on more complex datasets like CIFAR-10 or CelebA, which are standard benchmarks for diffusion models [1], are necessary to generalize the proposed metric. Examples from more complex models, such as StableDiffusion, should also be included.
   - *Robust experiments:* The authors select dataset-dependent ranges of timesteps to calculate variance based on prior knowledge, which may not be robust. They should provide principles or guidelines for selecting the range of timesteps for different datasets.

3. **Insufficient analysis in some areas.**
   - *Sub-approximation of score function leading to mode interpolation:* The claim that mode interpolation is caused by the inability of deep networks to learn ground truth score functions with sharp jumps for small $t$ needs support. A "sanity check" experiment using ground truth score functions is necessary but missing, making the claim less convincing. The authors should include this experiment and analyze the approximated score functions in cases with varying data sizes, as in Figure 2.
   - *Relationship between hallucinations and high variance along the trajectory:* The authors argue that hallucinated samples exhibit high variance along the denoising trajectory, as observed in Figure 5. This appears more empirical than intuitive. More discussion is needed to explain why hallucinated samples have high variance along the trajectory.

4. **Lack of formal definition for Eq. (4).**
    The definition of $\texttt{Hal}(x)$ in Eq. (4) is informal (and inconsistent or incorrect): (a) The summation of $i$ over $[0,T]$ contradicts the range used in experiments; (b) The calculation of $\overline{\hat{x}_0^{(t)}}$ is unclear, and the superscript $^{(t)}$ may be incorrect; (c) There is a typo where $i$ should be replaced by $t$. The authors should provide a formal definition for $\texttt{Hal}(x)$ to make the metric more rigorous.

[1] Denoising Diffusion Probabilistic Models, NeurIPS 2020.

**Questions:**

1. **How do studies on properties of mode interpolation relate to its cause or detection?**
   In Section 4, the authors discuss properties of mode interpolation, such as its occurrence between nearby modes and the effects of data size and denoising timesteps. How do these properties relate to the cause or detection of mode interpolation? Could the authors provide more insights into the cause of mode interpolation and how these properties are related?

2. **How can a threshold for $\texttt{Hal}(x)$ be determined?**
   In Section 5, the authors propose $\texttt{Hal}(x)$ as a metric to detect hallucinations. What value of $\texttt{Hal}(x)$ is considered a hallucination in the experiments? Could the authors provide more insights into determining the threshold for $\texttt{Hal}(x)$?

3. **Does the sampling algorithm affect hallucinations?**
   The sampling algorithm details are missing from the experiment settings. Could the authors provide more information on the sampling algorithm used in the experiments? Additionally, do different sampling algorithms (e.g., deterministic vs. stochastic, as discussed in [1]) affect hallucinations?

[1] Elucidating the Design Space of Diffusion-Based Generative Models, NeurIPS 2022.

**Limitations:**

Yes.

---

> ### Author Rebuttal · Authors · 2024-08-07
>
> Thank you for your thoughtful and detailed review. We are glad that you found our exploration of hallucinations in diffusion models and our explanation through mode interpolation to be novel and plausible. Your appreciation of our toy experiments validating the phenomenon is encouraging. We acknowledge your concerns and attempt to respond to them line by line below:
>
> ### **Re: Experiments on complex, and or natural datasets**
>
> Thank you for your comments on W1, W2.1. We kindly refer you to the [global response here](https://openreview.net/forum?id=aNTnHBkw4T&noteId=2W1mcxDdVO) along with the figures in the **attached PDF** for an exciting update with results on the Hands-11k dataset (and the extra finger problem)!
>
>
> ### **Re: Comprehensive and Robust Experiments**
>
> > **Sanity Check with Ground Truth Scores**:
>
> Thank you for pointing this out. We have done a sanity check experiment where we sampled using the ground truth score instead of the learned score function. We did not observe any hallucinations in this experiment. We included this in the Figure 3 (4th column) of the **attached PDF**. The analysis of approximated score functions across varying data sizes is something that we will definitely consider including in the final revision.
>
> ### **Re: Guidelines for selecting the range of timesteps for different datasets.**
>
> Our methodology for the selection of timesteps for any dataset was as follows. We plot the trajectory of the predicted x0 across various timesteps and find the region where x0 varies quite significantly. This gave us a good starting point for the selection of timesteps. For image datasets (shapes and Hands), we observe that similar timesteps ( t = 700 to t = 800) is a great starting point.
>
> ### **Re: Decoder’s Role: Relationship Between Hallucinations and High Variance**
>
> We provide a more detailed discussion on why hallucinated samples exhibit high variance along the denoising trajectory:
>
> The high variance in the trajectory of the hallucinated samples can be derived from the analysis of mode interpolation. The neural network learns a smooth approximation of the score function. The score function in diffusion models is (implicitly) learned to guide this reverse diffusion process. This smooth approximation leads to oscillations between the nearby modes which leads to hallucinations. Thus, we track the variance of the trajectory of x0 to detect the hallucinated samples. We hope this provides more intuition behind the proposed metric.
>
>
> ### **Re: Formal Definition of Eq. (4)**
> We sincerely apologize for this confusion and thank the reviewer for pointing it out. We define the corrected metric below (which was used in all of the experiments). The intuition remains the same, we want to capture the variance of predicted x0 in the reverse diffusion trajectory. Let $T_1$ be the start timestep and $T_2$ be the end timestep.
> Concisely, we can write it as
> $$
>  \text{Hal}(x) = \text{Var}(\hat{x_0} [T_1:T_2])
> $$
> In detail:
>
> $$
> \text{Hal}(x) = \text{Var}(\hat{x_0}[T_1:T_2]) = \frac{1}{|T_2 - T_1|} \sum_{i=T_1}^{T_2} \left( \hat{x_0}^{(i)} - \frac{1}{|T_2 - T_1|} \sum_{j=T_1}^{T_2} \hat{x_0}^{(j)} \right)^2
> $$
>
> ### **Re: Threshold for Hal(x)**
> The value of Hal(x) depends on the dataset and experimental setup. Depending on the approximate number of hallucinations, one can try to filter out the top x% of the generated samples. This should ideally retain most of the in-support samples and eliminate the hallucinated samples.
> ### **Re: Details of the Sampling Algorithm**
>
> We use the standard DDPM sampler with 250 steps while sampling for the Shapes setup. For the Gaussian experiments, we also used the standard DDPM sampler and the number of sampling steps was equal to the number of training steps (1000). We will update the details of the same in the paper as well.
>
>
> ---
>
> Please refer to the **attached PDF** for detailed results and figures from our additional experiments.
>
> We hope we were able to rest your concerns regarding the experimentation and convince you of its generality with the added results. We have made our best attempt at answering all the concerns raised, please do let us know if any other questions remain! Thank you for your concrete suggestions that ignited some nice experiments.

---

> > ### Comment · Reviewer_Zx7y · 2024-08-13
> >
> > Thanks for your response and the extensive additional experiments. Some of my concerns are addressed but some are not, listed below:
> > - **W1:** I think there exists a misunderstanding. What does "decoder" mean in the abstract and **Re4**? From the authors' supplementary PDF I guess it refers to "the upscaling stages of U-Net", but where is it mentioned in the paper?
> > - **W2.2 and Re3:** From the response, seemingly the range of t is selected from empirical case studies, giving non-convicing results. Still, I hold my opinion that $\mathrm{Hal}(x)$ should be calculated within a consistent and formal form to ensure generality, and the authors should discuss them as limitations or future works.
> > - **W3.2 and Re4:** The response about the connection between hallucination samples and high variances seems to be not enough convincing. What's the key to "smooth approximation leads to oscillations between the nearby modes"? In other words, why does not groundtruth score functions lead to oscillations between nearby modes?
> > - **Q1 and Re2:** Without providing supplemental experiment results (such as experiments across varying data sizes), could the authors give any intuitions behind how properties *(not limited to the data sizes, but including "occurrence between nearby modes" and "effects of denoising timesteps")* of mode interpolations, discribed in Section 4, relate to the cause or detection? Additionally, what's the difficulty for supplemental experiments across varying data sizes, and what's the authors' plan to conduct these experiments for the final version?
> > - **Q3 and Re6:** Thanks for your explanation, but I'm still concerned whether the sampling algorithm would affect the hallucinations. For example, [1] proposes that a stochastic sampler (like DDPM) can correct the under-estimation of score functions, compared with a deterministic sampler (like DDIM [2] with $\eta=0$). Could the authors provide results using DDIM samplers?
> >
> > [1] Elucidating the Design Space of Diffusion-Based Generative Models, NeurIPS 2022. \
> > [2] Denoising Diffusion Implicit Models, ICLR 2021.

---

> > > ### Author Response · Authors · 2024-08-14
> > >
> > > Thanks for the follow-up and continued engagement. We are happy to hear that some of your concerns were addressed with the original rebuttal, and we attempt to clarify the remaining questions below:
> > > - **W1**: Yes, the decoder refers to the upscaling stages of the U-Net.
> > > In the case of the 1D and 2D Gaussian experiments, since the input dimension is very small, there is no “encoder” as such. The 3-layer MLP acts as the decoder. We will improve the clarity of the term “decoder” in the revised version.
> > > - **W2.2 and Re3** : Thank you for this suggestion, we will include the limitations of the proposed method in the revised version. We want to highlight that the goal of this work was to identify and discover the phenomenon of mode interpolation, and its intricate relationship with hallucinations, and show promise in detecting this through a simple metric. We absolutely agree that future work should focus on developing improved, and informed metrics for detecting hallucinated samples, especially in real-world datasets.
> > > - **W3.2 and Re4**
> > >    - **Ground Truth Score Function:**
> > > We refer you to Figure 4, column 3 where we show how $\hat{x_0}$ is a smooth approximation of the step function (to show the connection between the learned score function, and its effect on the predicted $\hat{x_0}$).
> > > For instance, in the case of a mixture of Gaussians (1D), the score function precisely reflects the boundaries between different Gaussian components. This precision ensures that the score function is sharply defined in regions where the probability density changes abruptly, leading to no oscillations or artifacts between modes---informally, the predicted value snaps back to one of the modes, and is never in the region between modes because the force pulling it to the mode is so high. Essentially, the ground truth score function exactly mirrors the behavior of the true probability distribution, preventing any unintentional mixing or interpolation between modes.
> > >    - **Learned Score Function:**
> > > When the model generates samples, it relies on the learned (smoothed) score function to reverse the diffusion process. First, this is smooth and does not show the step-function-like behavior of the true score function. Since the learned score function cannot sharply separate the modes, it creates a smoother gradient between them, effectively leading to oscillations or interpolations between the modes---informally, creating a region of high variance/uncertainty where samples are being pulled to either mode with a high, but finite force. This is why samples can end up in regions where the ground truth distribution has low or even zero probability—these are the regions between the modes.
> > > - **Q1 and Re2**
> > >    - **The frequency of interpolated samples is inversely proportional to the number of sampling timesteps T’:** This is because if we have more timesteps, then the update on each $x_{t-1}$ given an $x_{t}$ would be smaller. This means that even when the $x_t$ is in the so-called region of uncertainty, it can quickly latch back to the nearest mode. On the contrary, if the sampling steps were less, each update means a larger step, leading to oscillations within the region of uncertainty, and from one mode to the other. Please note that there is a typo in the paper where we missed writing “inversely” proportional. The corresponding experimental results can be found in case of the VDM model in the **attached PDF**. We will include all of this and the updates in the final draft.
> > >    - **The number of interpolated samples also decreases as the distance from the modes increases:** Following the above explanation, if two modes are far apart, we need a larger shift from one  $x_{t}$ to the next $x_{t-1}$ to oscillate between modes. This once again means that the models can latch back to the existing mode much more easily.
> > >    - **As the number of training samples increases, we observe that the proportion of interpolated samples decreases:** This is primarily because more data enables to learn a better approximation of the score function.
> > >
> > > > Additionally, what's the difficulty for supplemental experiments across varying data sizes, and what's the authors' plan to conduct these experiments for the final version?
> > >
> > >  We note that we have run experiments with varying data sizes in Figure 9 of the paper. If the reviewer is asking about the detection results with varying data sizes, we plan to include the results of detection experiments with Gaussian1D and Gaussian2D across varying data sizes in the revised version. We agree that this would demonstrate the generality of the proposed detection metric.

---

> > > > ### Author Response · Authors · 2024-08-14
> > > > **New experiment with DDIM**
> > > >
> > > > **Q3 and Re6**: As requested, we ran a new experiment with DDIM sampler on the 1D-Gaussian dataset with 20k samples. We report the numbers below. While the sampler does have an impact on the frequency of hallucinations (in this case increasing it), the core phenomenon underlying hallucinations still persists. (Kindly note that the results below are is in the case of x-prediction)
> > > >
> > > > | Method | Fraction of Hallucinations (1e-4) |
> > > > |---|---|
> > > > | DDPM  | 22.35 |
> > > > | DDIM  | 40.59  |
> > > >
> > > > We appreciate your detailed feedback. Please let us know if you have any further questions.

---

### Official Review · Reviewer_8Ww6 · 2024-07-11

**Soundness:** 3
**Presentation:** 3
**Contribution:** 3
**Rating:** 6
**Confidence:** 4

**Summary:**

This paper demonstrates and studies a particular failure mode of diffusion models termed mode interpolation. Specifically, the authors discovered that when trained on certain datasets, diffusion models (even those with 1000 denoising steps) generate samples that look like certain interpolations of some training samples. The paper demonstrates the mode interpolation effect on several toy datasets (e.g., 1D and 2D Gaussians, grids) and the MNIST dataset.

The paper then delves into analyzing the cause of the mode interpolation behavior by examining the learned score functions. The authors observe that one cause of the artifact is that the denoising neural network cannot accurately mimic the score when it has abrupt changes.

The paper finally proposes a metric to estimate the plausibility of a sample being a mode-interpolated one based on the observation that “good” samples often have relatively small changes in the later sampling process.

**Strengths:**

The paper elaborates on a previously overlooked failure mode of diffusion models called mode interpolation. Specifically, when trained on certain datasets, diffusion models can generate samples that correspond to the interpolation of certain training samples. This phenomenon is not well-studied in prior work and this paper could bring further attention of the community to this problem.

The paper is also very well-written and easy to follow.

**Weaknesses:**

Despite the interesting observation of the mode interpolation behavior, my main concern is that the paper does not provide justifications for the mode interpolation behavior on more realistic datasets (e.g., natural images) and large models. While I totally understand that it is a major contribution to discovering the mode interpolation phenomenon, and it can be hard to observe this clearly on natural image datasets, I still believe some analysis can be done more carefully to let us understand mode interpolation better. I am happy to increase my score if the following concerns/questions are addressed.

In Figure 1, while the Gaussian example clearly indicates interpolated out-of-distribution samples, the SIMPLE SHAPES results do not seem to support the “interpolation” behavior. Specifically, by interpolating training images, we will always get blurry or gray shapes, but in all sampled images, the individual objects seem to be perfect in terms of both shape and color. It seems like the diffusion model is doing some sort of ``compositional generalization’’ over the training samples. While this is clearly problematic in this synthetic dataset, this could be a good behavior for diffusion models on natural images as they can generate unseen object combinations. This will not harm but actually improve the model. I think the case where diffusion models fail is when they generate “interpolation of training images”, in which case the image will be blurry or has other artifacts.

How do the denoising network structure and the forward noising schedule affect mode interpolation?

Looking closely at Figure 2, noting the log scale of the y-axis, it seems that a modest amount of samples (e.g., 50000) is sufficient to almost prevent mode interpolation from happening (1000 times less likely). I wonder will mode interpolation happen more often or less often when we increase the dimensionality to be similar to e.g. natural images.

In Section 4.3, the paper discusses the cause of the mode interpolation problem: the learned score function is inaccurate when the ground truth score changes significantly. It is still unclear how the sampling process interplays with this problem: does more steps mitigate or exaggerate the problem; and can other sampling methods (e.g., ODE-based) mitigate the mode interpolation problem?

In Section 5.2, can we apply a similar metric that does not require T sampled trajectories? Since in practice we only draw a few samples a time (e.g., per prompt).

**Questions:**

Please refer to the weakness section.

**Limitations:**

The authors discussed certain limitations in the appendix.

---

> ### Author Rebuttal · Authors · 2024-08-07
>
> We appreciate your review and are glad that you found our paper well-written & easy to follow, and that you appreciated our focus on the previously overlooked failure mode of diffusion models, namely mode interpolation. We acknowledge your concerns & attempt to respond to them line by line below:
>
> ### **Re: Results on Realistic datasets**
>
> Please refer to the [global response here](https://openreview.net/forum?id=aNTnHBkw4T&noteId=2W1mcxDdVO) along with the figures in the **attached PDF** for an exciting update with results on the Hands-11k dataset!
>
> ### **Re: Interpolation versus Compositional Generalization**
>
> We will answer this in two parts. First, we will distinguish between compositional generalization & interpolation. Second, we report the results of a concrete experiment to demonstrate how such interpolation happens in the embedding space.
> > **A. Why is hallucination different from compositional generalization?**
>
> The experimental design in our work is based on **unconditional** diffusion models, where the goal is to model the true p(x) of the distribution. The only way we interact with the learned distribution q(x) of the diffusion model is by sampling a random seed. This seed sampling is "in-distribution" to what was seen during training. Hence, the outputs should also be “in-distribution”. When we go into the text-image case, the text samples may be "outside" the distribution of the training set (eg, horse riding a man), which justifies composition in the output space as well (by being OOD), unlike the setting we are positioned in.
> > **B. Why are outputs not blurry if this is actually an interpolation?**
>
> This is a great question that led to the addition of a fun new experiment **(see PDF)** in favor of clarity! The interpolation is not happening in the output space, but rather in the representation space. We performed a t-SNE visualization of the outputs of the bottleneck layer of the U-net used in the Simple Shapes experiment. Please refer to Figure 2 in the **attached PDF** for the visualization. Regions 1 & 3 in the representation space semantically correspond to the images where squares are at the top & bottom of the image respectively. At inference time, we can see a clear emergence of region 2 which is between regions 1 & 3 (interpolated), & contains two squares (hallucinations) at the top & bottom of the image. This experiment concretely confirms that interpolation happens in representation space.
>
> ### **Re: Denoising Network Structure & Noising Schedule**
>
> > **Network Structure**:
>
> We systematically study this question by analyzing the count of hallucinations with various hidden dimension sizes in the architecture, & see that the hallucination count increases as the dimensionality of the hidden space increases. We hypothesize that a larger decoder may require even more samples to prevent hallucinations, which we showed in the paper reduces as the samples increase.
>
> These results are on Gaussian 1D (with 3 modes at 1, 2, 3) with 20k training samples.
>
> |Hidden Dimension|Hallucinations (1e-5)|
> |-|-|
> |64|1.36|
> |128|2.34|
> |256|2.48|
> |384|1.06|
>
> > **Noising Schedule**:
>
> We explore “Cosine” learning rate schedule similar to Improved DDPM work. In this case, we scale the betas to be in the same range as the linear schedule. We also experiment with a Quadratic schedule. The cosine learning schedule seems to reduce the fraction of hallucinations. These results are on Gaussian 1D (with 3 modes at 1, 2, 3) with 20k training samples.
>
> |Schedule Type|Hallucinations (1e-5)|
> |-|-|
> |Linear|2.34|
> |Cosine|0.10|
> |Quadratic|0.46|
>
> ### **Re: Sample Complexity & High Dimensionality**
>
> 1. First, we note that many of the experiments we studied were in very simple scenarios, such as 3 modes in 1 dimension & 50k samples. Despite the large number of samples for the task complexity, we did see hallucinations in this simple setup. In contrast, natural image datasets lie in a complex manifold with many modes. The example of missing/additional fingers in StableDiffusion (and are additional results on Hands-11K) shows that hallucinations persist with natural image datasets (and are much more frequent). Despite being trained on millions of natural images, the Stable Diffusion model fails to generate images of hands correctly. We believe that since the number of modes in real data are so many, the total number of samples required to prevent hallucinations is also much larger.
>
> 2. Second, we also know that real-world datasets are long-tailed in nature, so it is incredibly difficult to obtain a large number of samples to cover all the settings. One hypothesis in literature is that hands cover a small portion of the image & are often occluded in the images (for e.g by the person holding something). This long-tailed nature of real world datasets, makes hallucinations even more prevalent.
>
> ### **Re: Sampling Process & Hallucination Metric**
>
> We experimented with different sampling steps & found that increasing the number of sampling steps can reduce hallucinations. We refer the reviewer to Figure 3 of the **attached PDF** where we show this with Variational Diffusion Models. The first column (in Figure 3) with 250 sampling timesteps (T’ = 250) has more hallucinations compared to the second column with 500 sampling timesteps (T’ = 500).
>
> ### **Metric clarification**
>
> We believe that there is some confusion in understanding the metric in Eq.4. Here, t refers to the time-step during the reverse diffusion trajectory. Hence, we compute the variance of x0^ across select timesteps in a **single trajectory.** Hope this clarifies.
>
>
> ---
> Once again, we thank you for the constructive feedback on our work. We hope we were able to clarify all your concerns, and look forward to resolving any remaining concerns during the discussion phase.
>
> Please refer to the **attached PDF** for detailed results and figures from our additional experiments.

---

> > ### Comment · Reviewer_8Ww6 · 2024-08-12
> >
> > I thank the authors for their detailed response, which addresses many of my original concerns. In particular, I like the additional experiment that answers the question "why are outputs not blurry if this is actually an interpolation". It would be very helpful for the authors to incorporate these changes into the next version of their paper. After the rebuttal, I found the paper to have more merits than weaknesses so I will raise my score to 6.
> >
> > However, as also pointed out by the other reviewers, the fact that the paper lacks comprehensive experiments on real-world datasets is a major weakness. Specifically, evidence of interpolation is only observed in human-hand-related datasets. While I understand it is harder to find such effects in real-world cases, it might suggest that mode interpolation might "not really be a problem". For example, the authors observed mode interpolation in the latent space, which led to the perfect individual shapes in the SIMPLE SHAPES experiments. Given such results, one possible explanation for the unseen interpolation effect may come from the fact that the latent space is much more semantically compact compared to the pixel space and thus does not lead to "weirdly interpolated" images.
> >
> > But anyway, I think it is a solid contribution to observe the interpolation effect, even if it happens in the latent space in large diffusion models. This can inspire further thoughts and improve our understanding of diffusion models.

---

### Official Review · Reviewer_B7Kp · 2024-07-14

**Soundness:** 2
**Presentation:** 3
**Contribution:** 3
**Rating:** 5
**Confidence:** 4

**Summary:**

This paper studies the hallucination phenomenon in diffusion models, in which samples out of the support sets are generated. Specifically, the authors characterize a failure mode, termed mode interpolation, which is hypothesized to be attributed to the learned score function of the diffusion model being over-smoothed around the discontinuous jumps in the ground-truth score. The authors provide evidence for this hypothesis in experiments with 1D and 2D Gaussians, where the ground-truth scores are known. They then discovered the x0 predicted by the DDPM has higher variances (along the reverse diffusion trajectory) when generating hallucinated samples. Using this variance as a metric, the authors further show that they can filter out some hallucinated samples in datasets including Gaussians, SimpleShapes, and MNIST. They also demonstrate an application of such removal in recursive generative modeling, where samples generated by the current model are used to fine-tune the model.

**Strengths:**

1. The hallucination problems of diffusion models are less explored than those with LLMs. This is an interesting research direction. It is interaction with another research direction, recursive generative modeling, as highlighted by the authors, is also non-trivial.

2. I like the authors' thought process in approaching this question. Regions between modes are indeed a reasonable starting point for investigating a likelihood-based model.

3. The proposed metric for detecting hallucination appears to be effective in the three synthetic datasets.

**Weaknesses:**

1. It looks like only a particular parametrization of a particular type of diffusion model is tested, see Questions.

2. None of the datasets are natural images. Although the authors alluded to the analogy between mis-combining shapes and hands with 6 figures, the latter was not checked with real experiments. Would it be possible for the authors to apply the proposed hallucination metric to the hand generation problem and report some preliminary results?

**Questions:**

1. Though the over-smoothed score function is shown in Fig. 4, it is unclear if this phenomenon is general. From what I can gather, the authors only work with DDPM, which by design is not a rigorous likelihood-based model. The weighting between noise levels is adjusted to promote perceptual quality, compromising likelihood estimate. I request the authors to try some more rigorous likelihood-based diffusion models, e.g. Variational Diffusion Models, to further verify the generality of their discovery.

2. Even with DDPM, there can be different types of model parametrization, such that the neural networks are learning with different targets. The default setting for model parametrization in the GitHub repo is epsilon prediction, which naturally has higher variance around low noise levels (i.e. timesteps close to 0). Did authors try x-prediction and v-prediction, two popular alternatives to eps-prediction? Specifically, variances in the supervision targets of the neural network should be low around low noise levels for x-prediction, and static for v-prediction. I believe these ablation studies can help further justify the generality of their discovery.

**Limitations:**

I don't think the limitation of the proposed hallucination metric is sufficiently discussed. But I believe there should be negative societal impacts.

---

> ### Author Rebuttal · Authors · 2024-08-07
>
> Thank you for your thoughtful review. We are glad that you found our research direction intriguing (i) in exploring the hallucination problems of diffusion models, (ii) appreciated our approach of investigating regions between modes, and (iii) found our proposed metric effective for detecting hallucinations. We acknowledge your concerns and attempt to respond to them line by line below:
>
> ### **Re: Testing on Natural Images**
> We have discussed this point in the global response in detail and added new results on the Hands-11K dataset. Please refer to the [global response here](https://openreview.net/forum?id=aNTnHBkw4T&noteId=2W1mcxDdVO) along with the figures in the **attached PDF** for an exciting update.
>
> ### **Re: Testing other Models/Parametrizations**
>
> > **Variational Diffusion Models (VDM)**:
>
> We have conducted additional experiments using Variational Diffusion Models (VDM) to verify the generality of our findings based on your suggestion. Our results show that the over-smoothed score function phenomenon persists in VDM, supporting the hypothesis that this issue is not specific to DDPM.
>
> We train a simple VDM on the 2D Gaussian with 10k samples. We follow the setup and hyperparameters in the official implementation. We train both continuous and discrete variants of VDM on the 2D Gaussian dataset. We kindly refer the reviewer to the figure in the **attached PDF (Figure 3)**. The main observation is that VDM mitigates the hallucinations significantly especially with more training data but the phenomenon of mode interpolation still exists. In this figure, we also show the impact of the number of sampling steps on the count of hallucinations. We clearly see that increasing the number of sampling steps reduces the number of hallucinated samples. This can be clearly observed in Figure 2 (first two figures) where the count of hallucinations decreases mode interpolation.
>
>
> > **Alternative Parametrization**:
>
> Thank you for this suggestion. We have now explored different types of model parameterizations, including x-prediction and v-prediction, to understand their impact on hallucination detection. We observe that X-prediction is particularly worse in terms of the fraction of hallucinations. These results are on Gaussian 1D (with 3 modes at 1, 2, 3) with 20k training samples.
>
> | Method    	| Fraction of Hallucinations (1e-5) |
> |---------------|----------------------------------|
> | Eps-prediction  | 2.34                    	|
> | V-prediction  | 2.43                         	|
> | X-prediction  | 22.35                        	|
>
>
>
> ### **Re: Negative Societal Impacts and Limitations**
>
>
> Thank you for the nudge. We will add a detail on the limitations and societal impact of this work:
>
> In current text-to-image generative models, the poorly modeled “hands” are a clear giveaway in identification of AI generated images. The detection of such AI-generated content would be made much more difficult if these hallucinations were identified and removed from the generated images. While our work builds an understanding of hallucinations, and allows us to also detect them, we believe that future generations of models would have become more robust to such hallucinations by virtue of training on more data independent of this work.
>
> Concerning the limitations of the proposed hallucination metric, the selection of the right timesteps is key to be able to detect hallucinations. More analysis on what region of trajectory leads to hallucinations would be useful across various schedules and sampling algorithms. We believe these are great areas for future work to explore.
>
> Once again, we thank you for the constructive feedback on our work. Working on the pointers has helped us improve the quality of our analysis. We hope we were able to clarify all your concerns, and look forward to resolving any remaining concerns during the discussion phase.
>
> ---
>
> Please refer to the **attached PDF** for detailed results and figures from our additional experiments.

---

> > ### Comment · Reviewer_B7Kp · 2024-08-13
> >
> > Thank the authors for their detailed response. The results of VDM are particularly interesting and I hope the authors would like to make sure they are sufficiently discussed in the revised version. Given the observation that more sampling steps leads to less hallucinations in VDM, it seems like a continuous time VDM may be a well founded solution to reduce hallucination, given its theoretical grounding in being a decent likelihood estimate. I am increasing the rating but will still keep it borderline.

---

### Author Rebuttal · Authors · 2024-08-07

We appreciate the constructive feedback provided by all reviewers towards this submission. Across the board, all reviewers found the phenomenon of hallucination via mode interpolation as an interesting scientific inquiry and appreciated the quality of the draft that was supported with convincing, comprehensive, and rigorous experimental protocol.

There are a few common themes around weaknesses that all reviewers identified, which if acted upon could improve the draft. We took this feedback into strong consideration, and are excited to share the updated results, which substantially improve the significance and soundness of the results. Focusing on the weaknesses below:

### **Experiments on Real World Datasets**

This concern was raised by all reviewers. We understand the interest in seeing how our evaluation generalizes to natural image datasets. Following the general feedback, we have extended our experiments to include the hand generation problem to evaluate the effectiveness of our hallucination metric on real-world data. Specifically, we applied our proposed metric to the Hands-11k dataset [1] and observed instances of hallucinated samples with an incorrect number of fingers.

The Hands dataset [1] consists of high-resolution images of hands in various orientations. We sample 5000 images from the Hands dataset and train an ADM [2] model on this dataset. We resize the images to 128x128 and use the same hyperparameters as that of the FFHQ dataset (We mention the exact hyperparameters towards the end).
- We observe images with additional and missing fingers in the generated samples. This is **attached in the PDF** document.
- To analyze the effectiveness of the proposed metric, we manually label ~130 images from the generated samples as hallucinated vs. in-support. This includes ~90 images with 5 fingers and ~40 images with missing/ additional fingers i.e. hallucinated samples.
- The histogram (in the PDF) shows that the proposed metric can indeed detect these hallucinations to a reasonable degree. In our experiments, we observe that we can eliminate ~80% of the hallucinated samples while retaining ~81% of the in-support samples.
- We note that the detection is a hard problem and the fact that the method transfers to the real world is proof of the relationship between mode interpolation and hallucination in real-world data.

Please refer to the detailed results and corresponding figures in the **attached PDF (Figure 1)**. These results indicate that our metric is effective in detecting hallucinated samples in natural images as well. This also solidifies the connection between mode interpolation and hallucination in real-world datasets. This connects how additional fingers in StableDiffusion-generated images are closely linked to the ideas discussed in the paper.

### **Experiments on Alternative Parametrizations**

While the majority of the experiments in the submission were performed on DDPM models, we were nudged to expand the experiments to other parametrizations. We have added experiments with ADM model and likelihood-based Diffusion Models like Variational Diffusion Models in the rebuttal response.

—--------------

### Additional experimental details on the Hands dataset.

We trained for a total of 200k iterations with batch size 16 and a learning rate of 1e-4. The diffusion process was trained with 1000 steps with a cosine noise schedule. The U-Net comprised 256 channels, with an attention mechanism incorporating 64 channels per head and 3 residual blocks. For sampling, we use 500 timesteps with respacing.


[1] Afifi, M. (2019). 11K Hands: Gender recognition and biometric identification using a large dataset of hand images. Multimedia Tools and Applications. https://doi.org/10.1007/s11042-019-7424-8

[2] Nichol, Alexander Quinn, and Prafulla Dhariwal. "Improved denoising diffusion probabilistic models." International conference on machine learning. PMLR, 2021.

---

### Decision · Program_Chairs · 2024-09-25

**Decision:**

Accept (poster)

**Comment:**

This paper identifies a particular failure mode of diffusion models: "mode interpolation", which happens when diffusion models generate samples that look like certain interpolations of some training samples. The paper demonstrates the mode interpolation effect on several toy datasets (e.g., 1D and 2D Gaussians, grids, and MNIST), and later added the more realistic Hands dataset. After identifying the effect, the paper goes on to analyze the cause of the mode interpolation behavior by examining the learned score functions. The authors observe that one cause of the artifact is that the denoising neural network cannot accurately mimic the score when it has abrupt changes. After that, they propose a metric to estimate the plausibility of a sample being a mode-interpolated one based on the observation that “good” samples often have relatively small changes in the later sampling process.

Overall the paper has a really clear narrative, gave empirical evidence and made technical progress along the way, and the writing is clear and easy to follow.

Reviewers' main concerns center around the lack of real world dataset evaluation, which the authors later added in the rebuttal, the limitation on the model structure, which the author also complemented, and the suspicion that over-smoothed score function phenomenon is not general enough. Nonetheless, I think the identified phenomenon, the empirical evidence, and the prosed metric are all great contributions to the community, and the added dataset and model variant during rebuttal really strengthened the paper. I therefore suggest accept.